# Leg and hip muscles show muscle-specific effects of ageing and sport on muscle volume and fat fraction in male Masters athletes

Jochen Zange[1] , Joachim Endres[1], Christoph S. Clemen[1,2] and Jörn Rittweger[1,3]

[1]*Department of Muscle and Bone Metabolism, German Aerospace Centre (DLR), Institute of Aerospace Medicine, Cologne, Germany*
[2]*Department of Muscle and Bone Metabolism, Institute of Vegetative Physiology, Medical Faculty, University of Cologne, Cologne, Germany*
[3]*Department of Paediatrics and Adolescent Medicine, University Hospital of Cologne, Cologne, Germany*

Handling Editors: Karyn Hamilton & Christoph Centner

The peer review history is available in the Supporting Information section of this article (https://doi.org/10.1113/JP285665#support-information-section).

**Abstract figure legend** Masters athletes were compared with young athletes as well as old and young control subjects in terms of the volume and fat fraction of 17 different hip and leg muscles using a six-point DIXON magnetic resonance imaging sequence. Furthermore the peak power was determined during a countermovement jump. The sum of the jump muscles shown here includes the glutei, quadriceps and triceps surae muscles. The Masters athletes and the old controls exhibit a significantly lower jumping power compared to the younger comparison groups, which results from a smaller muscle volume (sarcopenia) to a small extent from a larger fat fraction (myosteatosis) and from volume-independent neuromuscular factors (sarcosthenia).

**Jochen Zange** is currently a senior scientist at the Institute of Aerospace Medicine at the German Aerospace Centre (DLR) in Cologne, Germany, and was awarded the venia legendi ('Privat-Dozent') for human physiology at the Medical Faculty of the University of Cologne. Dr Zange has conducted several studies in the field of applied physiology and pathophysiology of human skeletal muscle. Specifically using non-invasive methods he has focused on the loss of muscle volume and performance as well as adaptations in energy metabolism in astronauts and immobilized healthy subjects (bed rest studies), in ageing subjects and in patients with various neuromuscular disorders. In addition he has focused on the development and testing of gravity-independent training measures for astronauts and the application of nutrition and exercise therapies for patients.

**Abstract** Age-related deterioration in muscle volume, intramuscular fat content and muscle function can be modulated by physical activity. We explored whether Masters athletes, as examples of highly physically active people into old age, could prevent these age-related muscle deteriorations. Four groups of 43 men were examined: young athletes (20–35 years, $n = 10$), Masters athletes (60–75 years, $n = 10$) and two age-matched control groups (old: $n = 11$, young: $n = 12$). Volumes and fat fractions of 17 different hip and leg muscles were determined using magnetic resonance imaging. In the soleus muscle extra- and intramyocellular lipids were measured using 1H-MR-spectroscopy. Finally volumes of glutei, quadriceps and triceps surae muscles were cumulated and compared to peak jumping power. In both age groups the sum of glutei, quadriceps and triceps surae muscles showed larger volumes in athletes (young: $5758 \pm 1139$ ml, old: $5285 \pm 895$ ml) compared to the corresponding control groups (young: $4781 \pm 833$ ml, old: $4379 \pm 612$ ml) ($p < 0.001$). Fat fraction varied between 1.5% and 12.5% [1]H-signal across muscles and groups and was greater in Masters athletes than in young athletes ($p < 0.001$), but lower than that in old controls ($p < 0.001$) and comparable with young controls. Age and exercise-related effects on muscle fat predominantly originated from the extramyocellular lipids. Finally muscle peak power per volume was effectively halved in the combined older groups compared to the younger groups. Our findings suggest that sarcosthenia, that is, intrinsic muscle weakness, is an effective cause of age-related power declines in addition to sarcopenia and fat accumulation.

(Received 18 June 2024; accepted after revision 3 March 2025; first published online 28 March 2025)

**Corresponding author** J. Zange, German Aerospace Centre (DLR), Institute of Aerospace Medicine, Linder Höhe, 51147 Köln, Germany. Email: jochen.zange@dlr.de

**Key points**

- Muscle volume and muscle fat fraction from 17 hip and leg muscles of Masters athletes were compared with old controls, young athletes and young controls.
- Muscle volume and fat fraction were determined using magnetic resonance imaging (MRI) using a six-point-DIXON sequence.
- Muscle volume in Masters athletes was larger than that in old controls but partially smaller than that in young athletes.
- Muscle fat fraction of Masters athletes was lower than that in old controls but higher than that in young athletes.
- Muscles of old athletes and old controls produce only 50% of jumping peak power per muscle volume compared with younger subjects. The intrinsic reduction of power loss in old muscle could not be explained by the higher fat fraction in old muscle.

## Introduction

Sarcopenia is defined as the physiological, age-related loss in skeletal muscle volume (Rosenberg, 1989; Roubenoff & Hughes, 2000; Roubenoff et al., 1997); dynapenia (Clark & Manini, 2008) describes the age-related decline in muscle's mechanical output at the organismic level and sarcosthenia (Tanaka et al., 2019) at the material level. In old age significant alterations were found in nearly all examined mechanisms controlling muscle growth and muscle differentiation and therefore sarcopenia is commonly seen as a multifactorial process (Axelrod et al., 2023; Haran et al., 2012; Walston, 2012). Sarcopenia is traditionally being associated with an overall loss in muscle mass (Janssen et al., 2000). In the lower body it

appears that the age-related loss of muscle volume mainly affects the thigh muscles, whereas the loss in the hip muscles and lower leg muscles is less pronounced (Fuchs et al., 2023).

From bed rest immobilization studies we also know that hip and leg muscle are differently affected by atrophy. For example mobilizing muscles like the quadriceps muscles or the triceps surae muscles reacted with higher rates of volume loss than stabilizing muscles like the sartorius muscle (Belavy et al., 2009). Bed rest also induced different degrees of atrophy among different lumbar back muscles concerning volume loss and the increase in fat fraction (de Martino et al., 2022). When physical activity declines in old people, the mechanisms of immobilization-related

atrophy and sarcopenia overlap. Conversely it is well documented that old people can also gain muscle mass and strength through physical training (Carmeli et al., 2000; Marcell, 2003; Narici et al., 2004).

Recent research further suggests concomitant accumulation of adipocytes in the ageing muscle, referred to as myosteatosis (Correa-de-Araujo et al., 2020). The latter condition can non-invasively be measured as muscular fat fraction using magnetic resonance imaging (MRI) using a six-point DIXON sequence, which discriminates between the resonances of water protons and the protons in carbonyl groups (Grimm, Meyer et al., 2019). In addition an increased fraction of intra-muscular connective tissue has been reported (Csapo et al., 2014). Long-term inactivity, by 14 days (Fuchs, Hermans et al., 2025) or 60 days (de Martino et al., 2021, 2022) of experimental bed rest intervention, increased the fat fraction of skeletal muscles as assessed using MRI and tissue biopsies (Eggelbusch et al., 2024), whereas training interventions successfully reduced fat fraction in young (Grimm, Nickel et al., 2019) and older people (Flor-Rufino et al., 2023; Ghasemikaram et al., 2021).

Masters athletes are older people intensively pre-paring for and participating in competitive sports events, often at the national or international level. Therefore Masters athletes could potentially serve as a model for maximum effects of training as a countermeasure against sarcopenia. Some authors see Masters athletes as a model of successful ageing (Lazarus & Harridge, 2007; Rittweger et al., 2004). Thus evidence suggests that these persons may have conserved astonishingly high levels of exercise performance at advanced age, including much lower losses in strength, power and muscle volume, lower fat increase in muscle, a better conservation of neuromuscular control and enhanced markers of aerobic metabolism, compared with age-matched non-athletes (Drey et al., 2016; Hawkins et al., 2003; Joanisse et al., 2020; McKendry et al., 2018; Narici et al., 2004; Pollock et al., 2018). A recent case study of a 71-year-old female Masters athlete and world champion in powerlifting, who started strength training at the age of 63, showed that old people can not only maintain a high training status into old age but also become successful Masters athletes if they only start training in old age (Fuchs, Trommelen et al., 2024). The muscles of Masters athletes also adapt to the respective discipline. For example it has been observed that jumping performance is better maintained in sprint-trained Masters athletes than in endurance-trained Masters athletes (Michaelis et al., 2008). In addition Masters athletes who compete in sprint disciplines differ from endurance athletes in terms of their lean mass and the fat content of their bodies (Walker et al., 2023).

In this study we therefore selected only Masters athletes who participate in strength disciplines to analyse a homogeneous group in terms of muscle performance and body composition. It was determined to what extent they are spared from age-related muscle volume deficits and increased fat content. We further test the hypothesis that volume deficits and fat fraction increase by sarcopenia as well as the counteracting effects of athletics quantitatively differ between hip and leg muscles of different biomechanical function. Finally we estimated to what extent an age-related increase in muscle fat fraction explains a corresponding age-related loss in muscle power.

## Methods

### Ethical approval

The study was performed between September 2020 and May 2021 at the German Aerospace Centre in Cologne. The study conformed with the *Declaration of Helsinki* and received ethical approval from the Ärztekammer Nordrhein in Düsseldorf, Germany (approval no. 2018269), and was registered at the German clinical trials register (http://www.drks.de, registration no. DRKS00015764). Written informed consent was received from every participant.

In total 43 men were recruited and assigned to four groups. The originally planned 48 participants, that is, 12 per group, were unfortunately not reachable due to the COVID-19 pandemic. Two groups represented young athletes ($n = 10$, 20–35 years) and Masters athletes ($n = 10$, 60–75 years), respectively. The main inclusion criterion for athletes was regular training for and participation in national or international track and field competitions in sprinting or jumping events. The main inclusion criterion for non-physically active controls was reporting $\leq 25$ metabolic equivalents for task (MET) of physical activity per week. Two age-matched control groups included 12 young and 11 older non-athletic sub-jects, who did not participate in any regular sport or exercise. See our previous publications (Sanchez-Trigo et al., 2022; Scorcelletti et al., 2023) for more details on the results of the questionnaires on performance in the jumping and sprinting disciplines and the activity in metabolic unit per week, as well as the test results in this study on performance in the countermovement jump and forefoot hops. Following exclusion criteria were applied: smoking, diabetes mellitus (fasted serum glucose levels), a history of cardiovascular disease or any other condition that could have impact on the musculoskeletal system.

In average subjects of all four groups did not significantly differ in body size and mass (Table 1).

Magnetic resonance imaging (MRI) and spectroscopy were both performed in a 3 Tesla Siemens Biograph mMR system. In the magnet subjects were made to lay supine with their feet forward. The position of both feet was fixated by a foot holder. For MRI four body coils and a

**Table 1. Anthropometric data (means ± SD) of the cohort (Scorcelletti et al., 2023) of male subjects**

|                 | Young athletes | Young controls | Old athletes | Old controls |
|-----------------|----------------|----------------|--------------|--------------|
| Number          | 10             | 12             | 10           | 11           |
| Age (years)     | 24 ± 2         | 29 ± 4         | 65 ± 4       | 67 ± 5       |
| Height (cm)     | 180.2 ± 8.3    | 181.1 ± 6.5    | 177.6 ± 7.6  | 176.9 ± 5.6  |
| Body mass (kg)  | 77.4 ± 14.0    | 73.6 ± 13.0    | 84.8 ± 33.8  | 79.8 ± 8.9   |

All four groups were matched for height ($p = 0.591$) and body mass ($p = 0.655$). Age was matched within old subjects ($p = 0.399$), whereas young controls were slightly older than young athletes ($p = 0.005$).

spine coil were used. Five sets of 52 continuous, trans-axial images were recorded reaching from the feet to the lumbar spine or at minimum to the upper edge of the entire pelvis, depending on the body size. The images focused on the right of the hip and the right leg. The field of view was 300 × 300 mm with a slice thickness of 5 mm. The acquisition matrix was 256 × 256 with a resolution of 1.17 × 1.17 × 5.0 mm per voxel. A turbo six-echo-DIXON sequence (Grimm et al., 2018b; Grimm, Nickel et al., 2019) with a flip angle of 5°; a repetition time of 10 ms; and echo times of 1.35, 2.64, 3.69, 5.22, 6.51 and 7.80 ms, respectively, were used. From this sequence images with grey values were received, representing the fat fraction given with a resolution from 0 to 1000. These values were read as fat fraction given in percentage of the total [1]H signal with a precision of 0.1%. The fat fraction represented the entire $-CH_2-$ signal, because the DIXON sequence does not distinguish between extramyocellular lipid (EMCL) and intramyocellular lipid (IMCL):

$$\text{Fat fraction (FF)} = 100 \times \frac{(-CH_2-)}{(-CH_2-) + (H_2O)} \%$$

$$\text{of the entire}^{1}\text{H signal}$$

Images of fat fraction were further analysed using the semi-automatic muscle segmentation module included in Mimics 25 (Materialise N.V., Leuven, Belgium) (Kolk et al., 2015). The muscle segmentation module identified all muscles in the hips and legs based on 10 atlases provided with the software. All muscles were shown as 3-D matrices. The location of each muscle matrix was manually corrected where needed. Typically, only some minor corrections were needed in some slices located in the hip showing the four origins of the quadriceps muscle. Muscle matrices of 17 selected muscles were exported as text files containing a table listing each voxel with the three coordinates and the fat fraction value, respectively. The 17 muscles included the 3 gluteus muscles (maximus, medius and minimus), the quadriceps muscle (m. vastus lateralis, medialis, intermedius, m. rectus femoris), the hamstrings (m. biceps femoris caput longus, m. biceps femoris caput breve, m. semimembranosus, m. semi-tendinosus), the m. sartorius and m. gracilis, the triceps

surae muscle (m. gastrocnemius medialis and lateralis, m. soleus) and finally the m. tibialis anterior. This selection included muscles running perpendicular to the body length axis and which therefore occurred in several of the trans-axial images being beneficial for the precision of segmentation. Moreover this selection included the major drivers for locomotion and some representants of joint stabilizers (m. sartorius, m. gracilis, m. tibialis anterior).

From each matrix file the corresponding muscle volume (ml) was calculated by counting all voxels and multiplying the voxel number and the voxel size. Further the average fat fraction (% of the total [1]H-signal) of the entire muscle was determined. Then each muscle matrix was split into a peripheral and a central region to find the regional differences in fat fraction. The periphery contained all voxels from the most proximal and the most distal muscle slices and an outer ring with a thickness of two voxels or 2.34 mm from the slices in between the end slices. All remaining voxels in the muscle matrix were counted as central voxels.

In addition the matrices of the glutei, quadriceps and triceps surae muscles were summed to determine how a reduction in functional muscle volume influences jumping power. From this composite matrix the total muscle volume and the mean fat fraction of the jumping muscles were calculated. The lean volume of the jumping muscles was estimated by multiplying the volume by the mean water fraction (100 – fat fraction).

We excluded three muscles from further evaluation, because these muscles contained on average between 25% and 40% fat. In the fat fraction images these muscles exhibited a clear contrast to the surrounding muscles containing much less fat. The three muscles were a gastro-cnemius medialis muscle from an old control subject and a biceps femoris longus and a semitendinosus muscle from two different old athletes. The affected subjects did not report any symptoms.

## Magnetic resonance spectroscopy

For spectroscopy a single flex coil was placed under and folded around the belly of the right calf muscle. [1]H-magnetic resonance spectroscopy ([1]H-MRS) was

performed in a single voxel of $15 \times 15 \times 15$ mm$^3$ placed in the belly of the right soleus muscle. The following acquisition parameters were applied: pulse angle 90°, repetition time 10,000 ms, echo time 30 ms, bandwidth 2000 Hz and two preparation scans followed by eight averaged scans. The integrals of the water signal and the carbonyl protons ($-CH_2-$) were evaluated from EMCL and IMCL (see Fig. 1) as described previously (Boesch & Kreis, 2000; Schick et al., 1993). The resonance frequencies of carbonyl protons differ because of the highly ordered structure of the skeletal muscle and the interaction between lipids stored in large volumes in adipocytes (EMCL) and their orientation in the magnetic field. The lipids of the muscle cell are located in small droplets and are not affected by a field orientation (Boesch & Kreis, 2000; Kreis & Boesch, 1994). The three $^1$H-signals were identified and quantified using the AMARES algorithm included in the free software jMRUI 5.2 (http://www.jmrui.eu) (Naressi et al., 2001; Stefan et al., 2009).

### Determination of lipids in biopsy samples of the soleus muscle

Biopsies were taken from the left soleus muscle of each subject, and parts of the tissue specimens were used to prepare 6 μm-thick cryosections, which were stained for a standard myopathological evaluation (to be published elsewhere), including oil red O for neutral lipid staining. The sizes of the oil red O droplets were divided into three categories: (1) small sized, (2) medium sized and (3) coarse sized.

### Determination of the peak power at a countermovement jump

All subjects performed a set of three countermovement jumps on a force plate (Leonardo Mechanography GRFP, Novotec Medical Inc., Pforzheim, Germany) continuously measuring the vertical ground reaction forces at a sampling rate of 800 Hz. Using the manufacturer's software the recorded force and the body mass determined at rest before the jump (weight/9.81 m $\times$ s$^{-2}$) were first used to calculate an acceleration value (ground force/body mass). From acceleration the speed of movement of the centre of mass was calculated using integration. The actual jumping power and the peak power reached before launch were then calculated from the product of the ground force and speed. For correlation with the volume of jumping muscles, the peak power (kW) was calculated as the median of the three measured values.

### Statistics

For statistical evaluation the software SPSS Statistics, version 26 (IBM-Deutschland GmbH, Böblingen, Germany), was used.

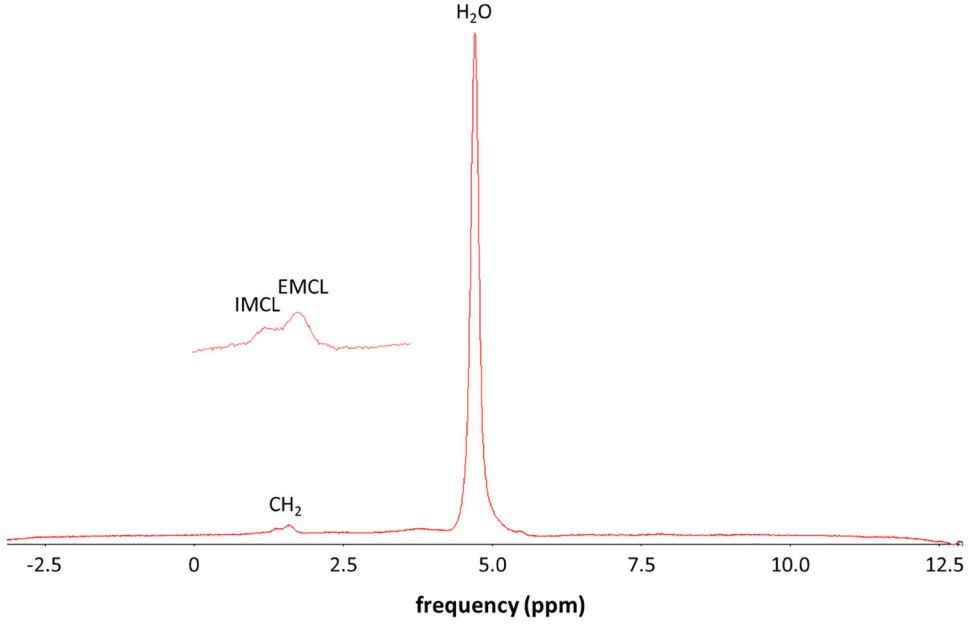

**Figure 1. Example of a $^1$H-MR spectrum acquired from the soleus muscle showing resonances from water (H$_2$O) and carbonyl (–CH$_2$–) protons within lipids**
The carbonyl resonance was split into two peaks referring to extramyocellular lipids (EMCLs) located in adipocytes and intramyocellular lipids (IMCLs) of muscle fibres (Boesch & Kreis, 2000). The frequency is given as a chemical shift (in ppm) from the pulse frequency. The amplitude is given in arbitrary units.

Linear mixed effect (LME) models were used to find group differences in muscle volume and in fat fraction. The aforementioned three individual muscles from different subjects that contained an unusually high fat content were excluded from the statistical analysis. In general prior to the LME test, data were tested for normal distribution using the Kolmogorov–Smirnov test and a q–q plot. Where necessary box-cox transformation of data was applied to obtain normal distribution. Finally the validity of the LME analysis was tested by proving the normal distribution of residuals as mentioned earlier.

In a first analysis data from all muscles were tested using the fixed factors *muscle* (the 17 muscles), *age* (old, young), *sport* (athletes, controls), *muscle\*age* and *muscle\*sport*, with *muscle* also being a repeated factor. This analysis included a Bonferroni-corrected univariate analysis of *muscle* differences in each test group.

In a second step LME was separately applied on each of the 17 muscles and on the combined muscle group of jumping musculatures, including glutei, quadriceps, hamstrings and triceps surae muscles. The variables volume and fat fraction were tested for the fixed factors *age* (old, young), *sport* (athletes, controls) and *age\*sport*. Bonferroni-corrected univariate analyses were included for a pair-wise testing of sport effects in each separate age group and of age effects in each sport group. Furthermore the effect size of pair-wise group differences was determined as Cohen's *d* that was calculated from the difference in averages and the common standard deviation.

For each muscle LME was further used to test for differences in fat fraction concerning the pixel *location* within the muscle matrix (periphery, centre) *versus* the four *AgeSport* test groups (old control, old athlete, young control, young athlete) again, including univariate sub-analyses with Bonferroni correction. For soleus muscle LME was finally applied to analyse the variables EMCL and IMCL from [1]H-MRS on *age* and *sport*, including the corresponding univariate subtests. Pearson's correlations between the summed volume of the jumping musculature and the peak power of countermovement jumping or between the corresponding lean volume and the peak power, respectively, were first calculated for all four subject groups. Because the correlations of control and athlete values matched within each age group, the final correlations were calculated for old and young subjects. ANOVA was used to test the corresponding models for significant differences. The oil red O staining of neutral lipid droplets in the biopsy samples was analysed using the Kruskal–Wallis test for independent samples.

The general level of significance was $p < 0.050$. The effect size Cohen's *d* is typically divided into the three categories: small: $d > 0.200$, medium: $d > 0.500$ and large: $d > 0.800$.

## Results

### Volumes of hip and leg muscles

Analysis of muscle volumes showed significant difference for the main factors *muscle* (df = 14/68.886, $p < 0.001$), *age* and *sport* (both, df = 1/198.943, $p < 0.001$) and also for age in different muscles (*age\*muscle*: df = 14/68.886, $p = 0.0113$) and sport in muscles (*sport\*muscle*: $p < 0.001$). We therefore decided to treat each muscle as an individual organ, which was separately tested for age and sport effects.

All examined muscles in both athlete groups had larger volumes than the corresponding muscles in the non-athletic controls (df = 39, $p$-values range from <0.001, e.g. for the rectus femoris muscle, to 0.0458 for the gastrocnemius lateralis muscle), except for the gluteus minimus muscle (df = 39, $p = 0.844$) (Fig. 2,

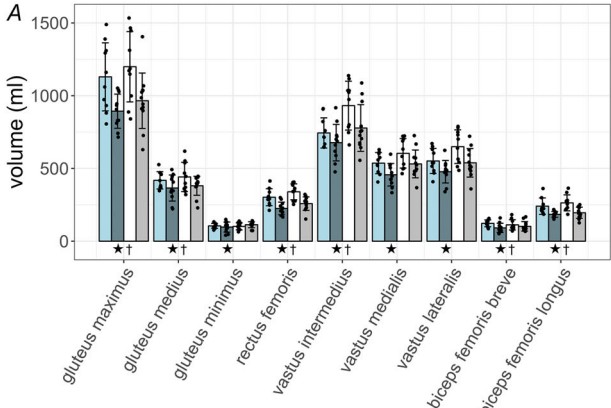

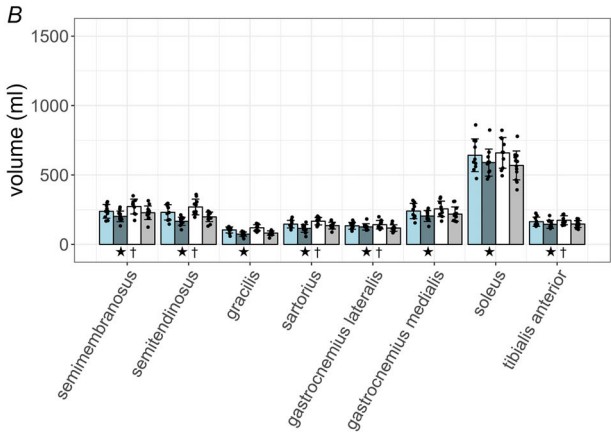

**Figure 2. Volumes (ml, mean ± SD, $n = 43$ or 42, all male, see Table 2) of hip and leg muscles**

Almost all muscles in athletes had higher volumes than those in controls. A lower volume by age was seen only in thigh muscles. †: significant effects by sport, ⋆: significant effects by age. For the sake of clarity, the figure was divided into two subfigures (*A* and *B*). The $p$-values for the effects of sport and age are presented in Table 2.

all $p$-values are presented in Table 2). A study of the two age groups separately showed that sport significantly affected 8 muscles in old and 11 muscles in young subjects, respectively. Among the old participants the significant effects of sport range from $p < 0.001$ and effect size $d = 0.562$ (df = 39) in the gracilis muscle to $p = 0.0443$ and $d = 0.310$ (df = 39) in the vastus medialis muscle. The significant effects of sport in the young group range from $p < 0.001$ and $d = 0.575$ in the semitendinosus muscle (df = 38) to $p = 0.0372$ and $d = 0.329$ in the tibialis anterior muscle (df = 39). All $p$- and $d$-values are presented in Table 2.

Age reduced the muscle volume only in about half of the muscles investigated here; for example, quadriceps muscles and hamstrings were significantly affected by ageing (df = 39, $p$-values range from 0.00924 for m. vastus intermedius to 0.0297 for m. rectus femoris), whereas the glutei muscles (df = 39, $p = 0.404$ m. gluteus minimus, $p = 0.437$ m. gluteus medius, $p = 0.544$ m. gluteus maximus) and the triceps surae muscles (df = 39, $p = 0.354$, m. gastrocnemius medialis, $p = 0.998$, gastrocnemius lateralis, $p = 0.962$ m. soleus) were not affected (Fig. 2, $p$-values for all muscles in Table 2). A study of the two sport groups showed that the small effects of age remained significant only for the volumes of the vastus lateralis muscle (df = 39, $p = 0.0135$, $d = 0.395$, Table 2) and the vastus intermedius muscle (df = 39, $p = 0.0261$, $d = 0.353$, Table 2), whereas in the controls the age effect on volume remained significant only for the vastus medialis muscle (df = 39, $p = 0.0492$, $d = 0.310$, Table 2).

Comparing pair-wise using $t$ tests the sum of the volumes of the composed muscle groups of Masters athletes with the three other groups, we found for the glutei muscles a significant difference of +18% *versus* old controls ($p = 0.0160$, $d = 1.200$) and no significant differences to young athletes ($p = 0.537$) and young controls ($p = 0.115$). The volume of the quadriceps muscle of Masters athletes differs by −18% from those of young athletes ($p = 0.0269$, $d = −1.093$) and by +14% from old control ($p = 0.0222$, $d = 1.091$), with no significant difference to young controls ($p = 0.840$). The volumes of hamstrings of Masters athletes differ by +20 from volumes of old controls ($p = 0.0132$, $d = 1.275$) and were not significantly different from young athletes ($p = 0.336$) and young controls ($p = 0.149$). The volume of the triceps surae muscle of Masters athletes did not differ significantly from the volumes of young athletes ($p = 0.691$), old control subjects ($p = 0.306$) and young control subjects ($p = 0.141$).

### Fat fraction in hip and leg muscles by six-point DIXON MRI

In the overall analysis of the 17 muscles, the fat fraction significantly differed between muscles (df = 14/78,409,

$p < 0.001$). Significant effects were also found for age and sport (both, df = 1/488,459, $p < 0.001$). Again all muscles were further analysed separately.

Values of the fat fraction from all examined muscle are shown in Fig. 3. The $p$-values of main effects and the $p$-values and the effect sizes of univariate test are summarized in Table 3.

The fattest muscles, namely the gluteus maximus muscle followed by the four heads of the hamstrings, were located proximal and dorsal (Fig. 3). More specifically among controls the fat fraction of all muscles was increased by age, with $p$-values ranging from <0.001, for example, in the gluteus maximus muscle, to 0.00907 for the sartorius muscle (df = 38 or 39) with medium to big effect sizes. Notably among athlete muscles biceps femoris caput breve (df = 39, $p = 0.190$), biceps femoris caput longus (df = 38, $p = 0.0564$), the sartorius (df = 39, $p = 0.0957$) and the gracilis (df = 39, $p = 0.169$) muscles were not affected by age. All other muscles exhibited an increased fat fraction in the Masters compared with the young athletes, with $p$-values ranging from 0.001 (e.g. m. soleus) to 0.0365 (m. gluteus medius) with small to big effect sizes. (df = 38 or 39, Fig. 3; Table 3). In the older subjects sport significantly reduced the fat fraction in 11 of 17 muscles, with $p$-values ranging from 0.001 (e.g. m. gracilis) to 0.0362 (m. tibialis anterior) with small to medium effect sizes (Fig. 3; Table 3). In the young subjects the fat fraction of six muscles was lower in the athletes than in the controls ranging from $p = 0.0095$ (m. gracilis) to $p = 0.0480$ (m. sartorius) with small effect sizes (Fig. 3; Table 3). Remarkably in both age groups the fat fractions of the entire vastus muscle and the semimembranosus muscle were not affected by sport (df = 39, $p$-values range from 0.0992, old m. vastus intermedius, to 0.314, old m. vastus medialis, Fig. 3; Table 3).

Comparing pair-wise using $t$ tests the mean fat fraction of the composed muscle groups of Masters athletes with the other three groups, the fat fraction of glutei of Masters athletes differs by +54% from young athletes ($p = 0.00953$, $d = 1.408$) and by −84% from old controls ($p = 0.00351$, $d = −1.443$), with no significant difference to young controls ($p = 0.577$). The fat fraction of quadriceps muscles of Masters athletes was 43% higher than that of young athletes ($p = 0.0046$, $d = 1.576$) and was not significantly different from that of young controls ($p = 0.0988$) and old controls ($p = 0.131$). Concerning the hamstrings of Masters athletes, fat fraction was +57% higher compared with young athletes ($p = 0.0490$, $d = 1.012$) and not significantly different from old controls ($p = 0.277$) and young controls ($p = 0.266$). In triceps surae muscle the fat fraction of Masters athletes was +51% higher than that in young athletes ($p = 0.00133$, $d = 1.966$), −34% lower than that in old controls ($p = 0.0392$, $d = −0.970$) and +33% higher than that in young controls ($p = 0.0242$, $d = 1.115$).

**Table 2. p-Values resulting from linear mixed effect models testing the *volume* of each muscle for the main effects age and sport and age*sport and from univariate retests with Bonferroni correction**

| | | | Main effects | | | | Univariate tests — Age in sport | | | | Univariate tests — Sport in age | | |
|---|---|---|---|---|---|---|---|---|---|---|---|---|---|
| | | | Age | Sport | Age*sport | Athlete | | Control | | Old | | Young | |
| Muscle | n | df | p | p | p | p | d | p | d | p | d | p | d |
| Gluteus maximus | 43 | 39 | 0.544 | 0.00190 | 0.862 | 0.593 | 0.082 | 0.751 | 0.049 | 0.0331 | 0.336 | 0.0165 | 0.382 |
| Gluteus medius | 43 | 39 | 0.437 | 0.0259 | 0.872 | 0.521 | 0.099 | 0.651 | 0.070 | 0.140 | 0.230 | 0.0845 | 0.270 |
| Gluteus minimus | 43 | 39 | 0.404 | 0.844 | 0.313 | 0.774 | 0.044 | 0.130 | 0.236 | 0.459 | 0.114 | 0.302 | 0.159 |
| Rectus femoris | 43 | 39 | 0.0297 | <0.001 | 0.889 | 0.109 | 0.250 | 0.129 | 0.236 | 0.00129 | 0.529 | <0.001 | 0.571 |
| Vastus intermedius | 43 | 39 | 0.00924 | 0.00294 | 0.525 | 0.0261 | 0.353 | 0.132 | 0.234 | 0.0842 | 0.270 | 0.00957 | 0.416 |
| Vastus medialis | 43 | 39 | 0.0117 | 0.00707 | 0.901 | 0.0929 | 0.263 | 0.0492 | 0.310 | 0.0443 | 0.317 | 0.0594 | 0.296 |
| Vastus lateralis | 43 | 39 | 0.00284 | 0.0131 | 0.553 | 0.0135 | 0.395 | 0.0654 | 0.289 | 0.169 | 0.214 | 0.0278 | 0.348 |
| Biceps femoris, caput breve | 43 | 39 | 0.987 | 0.0358 | 0.288 | 0.472 | 0.111 | 0.428 | 0.122 | 0.0283 | 0.347 | 0.438 | 0.120 |
| Biceps femoris, caput longus | 42 | 38 | 0.221 | <0.001 | 0.674 | 0.267 | 0.174 | 0.545 | 0.094 | 0.00662 | 0.443 | <0.001 | 0.560 |
| Semimembranosus | 43 | 39 | 0.0450 | 0.00819 | 0.774 | 0.115 | 0.246 | 0.199 | 0.199 | 0.0882 | 0.267 | 0.0340 | 0.335 |
| Semitendinosus | 42 | 38 | 0.01455 | <0.001 | 0.817 | 0.0668 | 0.291 | 0.0916 | 0.267 | 0.00255 | 0.498 | <0.001 | 0.575 |
| Gracilis | 43 | 39 | 0.0499 | <0.001 | 0.495 | 0.0709 | 0.283 | 0.335 | 0.149 | <0.001 | 0.562 | <0.001 | 0.724 |
| Sartorius | 43 | 39 | 0.0158 | <0.001 | 0.948 | 0.0843 | 0.270 | 0.0794 | 0.275 | 0.0118 | 0.403 | 0.00812 | 0.425 |
| Gastrocnemius lateralis | 43 | 39 | 0.998 | 0.0458 | 0.301 | 0.479 | 0.109 | 0.446 | 0.117 | 0.482 | 0.108 | 0.0322 | 0.339 |
| Gastrocnemius medialis | 42 | 38 | 0.354 | 0.024 | 0.957 | 0.496 | 0.106 | 0.526 | 0.099 | 0.119 | 0.246 | 0.089 | 0.269 |
| Soleus | 43 | 39 | 0.962 | 0.0357 | 0.571 | 0.722 | 0.055 | 0.652 | −0.069 | 0.268 | 0.171 | 0.0568 | 0.299 |
| Tibialis anterior | 43 | 39 | 0.500 | 0.0151 | 0.635 | 0.433 | 0.121 | 0.883 | 0.023 | 0.156 | 0.220 | 0.0372 | 0.329 |

The significance level was $p < 0.050$; significant values are in red.

Abbreviations: d, effect sizes for univariate tests given as Cohen's d; df, degrees of freedom; n, number of subjects.

**Table 3. p-Values resulting from linear mixed effect models testing the FF of each muscle on the fixed main effects of age and sport and age*sport and from univariate retests with Bonferroni correction**

| Muscle | n | df | Main effects — Age p | Main effects — Sport p | Main effects — Age*sport p | Age in sport — Athlete p | Age in sport — Athlete d | Age in sport — Control p | Age in sport — Control d | Sport in age — Old p | Sport in age — Old d | Sport in age — Young p | Sport in age — Young d |
|---|---|---|---|---|---|---|---|---|---|---|---|---|---|
| Gluteus maximus | 43 | 39 | <0.001 | 0.00218 | 0.564 | 0.00739 | 0.428 | <0.001 | 0.585 | 0.0101 | 0.407 | 0.0610 | 0.291 |
| Gluteus medius | 43 | 39 | <0.001 | 0.0438 | 0.764 | 0.00219 | 1.541 | 0.00388 | 1.429 | 0.220 | 0.189 | 0.0963 | 0.258 |
| Gluteus minimus | 43 | 39 | <0.001 | 0.0901 | 0.805 | 0.00365 | 0.472 | 0.00535 | 0.449 | 0.304 | 0.161 | 0.164 | 0.217 |
| Rectus femoris | 43 | 39 | <0.001 | 0.00929 | 0.387 | 0.0102 | 0.407 | <0.001 | 0.667 | 0.0156 | 0.373 | 0.191 | 0.203 |
| Vastus intermedius | 43 | 39 | <0.001 | 0.0408 | 0.768 | 0.00626 | 0.622 | 0.00108 | 0.747 | 0.0992 | 0.258 | 0.201 | 0.198 |
| Vastus medialis | 43 | 39 | <0.001 | 0.118 | 0.888 | 0.00122 | 0.532 | 0.00109 | 0.538 | 0.314 | 0.155 | 0.221 | 0.189 |
| Vastus lateralis | 43 | 39 | <0.001 | 0.152 | 0.634 | 0.0365 | 0.330 | 0.00438 | 0.461 | 0.182 | 0.207 | 0.488 | 0.107 |
| Biceps femoris, caput breve | 43 | 39 | <0.001 | <0.001 | 0.0382 | 0.190 | 0.203 | <0.001 | 0.697 | <0.001 | 0.724 | 0.0828 | 0.272 |
| Biceps femoris, caput longus | 42 | 38 | <0.001 | <0.001 | 0.267 | 0.0564 | 0.304 | <0.001 | 0.593 | 0.00103 | 0.549 | 0.0424 | 0.324 |
| Semimembranosus | 43 | 39 | <0.001 | 0.0873 | 0.998 | 0.00441 | 0.461 | 0.00248 | 0.493 | 0.227 | 0.187 | 0.217 | 0.191 |
| Semitendinosus | 42 | 38 | <0.001 | 0.002 | 0.842 | 0.0142 | 0.396 | 0.00338 | 0.482 | 0.0194 | 0.377 | 0.0290 | 0.350 |
| Gracilis | 43 | 39 | 0.00193 | <0.001 | 0.220 | 0.169 | 0.214 | 0.00192 | 0.508 | <0.001 | 0.674 | 0.0095 | 0.416 |
| Sartorius | 43 | 39 | 0.00338 | 0.00124 | 0.535 | 0.0957 | 0.260 | 0.00907 | 0.439 | 0.00648 | 0.439 | 0.0480 | 0.311 |
| Gastrocnemius lateralis | 43 | 39 | <0.001 | <0.001 | 0.432 | 0.00262 | 0.490 | <0.001 | 0.703 | 0.00412 | 0.465 | 0.0553 | 0.301 |
| Gastrocnemius medialis | 42 | 38 | <0.001 | <0.001 | 0.545 | 0.00222 | 0.506 | <0.001 | 0.665 | 0.00189 | 0.515 | 0.0131 | 0.402 |
| Soleus | 43 | 39 | <0.001 | 0.00708 | 0.366 | <0.001 | 0.618 | <0.001 | 0.867 | 0.0121 | 0.401 | 0.176 | 0.210 |
| Tibialis anterior | 43 | 39 | <0.001 | 0.00420 | 0.954 | <0.001 | 0.568 | <0.001 | 0.621 | 0.0362 | 0.331 | 0.0394 | 0.325 |

The significance level was p < 0.050; significant values are in red.

Abbreviations: d, effect sizes for univariate tests are given as Cohen's d; df, degrees of freedom; FF, fat fraction; n, number of subjects.

The separate analysis of voxels located in the periphery and the remaining voxels in the centre of each muscle showed that in general the outer voxels contained a higher fat fraction than the inner voxels (df = 76 or 78, $p < 0.001$). Furthermore in all muscles significant differences were found across the four groups (df = 76 or 78, $p < 0.001$, Tables 4 and 5). The univariate testing for local differences in fat fraction for each muscle confirmed this finding except for a few cases (Table 5).

## Intra- and extramyocellular fat fraction in the soleus muscle

[1]H-MR spectra were obtained from a $15 \times 15 \times 15$ mm[3] voxel of the right soleus muscle and exhibited a signal pattern in which the amount of EMCLs was always distinctly higher than that of IMCLs (Fig. 4).

EMCL exhibited significant main effects of age (df = 39, $p < 0.001$) and sport (df = 39, $p = 0.00328$, Fig. 4).

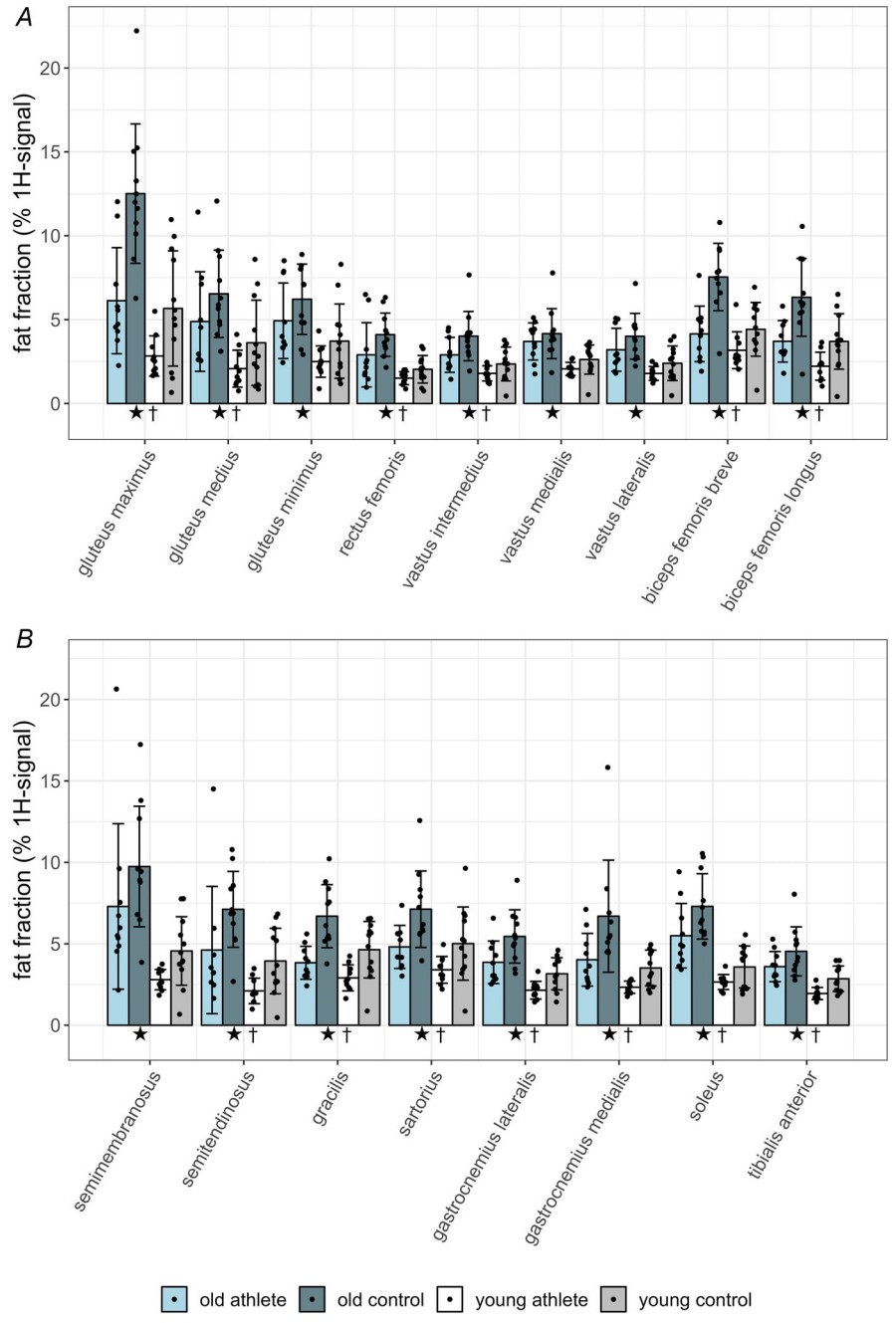

**Figure 3. Fat fraction (% of total [1]H-signal, mean ± SD, *n* = 43 or 42, all male, see Table 3) in selected hip and leg muscles measured using six-point DIXON MRI (magnetic resonance imaging)**

In all muscles the fat fraction was increased by age. In most muscles the fat fraction was lower in athletes than in age-matched controls. †: significant effects by sport, ★: significant effects by age. For the sake of clarity, the figure was divided into two subfigures (*A* and *B*). The *p*-values for the effects of sport and age are presented in Table 3.

**Table 4. Fat fraction (% of total $^1$H-signal, mean ± SD, n = 43 or n = 42, all male) in selected hip and leg muscles**

| Muscle | Old athletes | | | Young athletes | | | Old controls | | | Young controls | | |
|---|---|---|---|---|---|---|---|---|---|---|---|---|
| | Shell | Centre | Δ | Shell | Centre | Δ | Shell | Centre | Δ | Shell | Centre | Δ |
| Gluteus maximus | 15.3 ± 5.7 | 5.4 ± 3.0 | 9.9 | 8.9 ± 3.4 | 2.4 ± 1.0 | 6.5 | 27.9 ± 5.3 | 11.1 ± 4.2 | 16.8 | 15.0 ± 8.1 | 4.8 ± 3.1 | 10.2 |
| Gluteus medius | 9.9 ± 3.9 | 3.9 ± 2.8 | 5.9 | 5.0 ± 1.8 | 1.5 ± 1.0 | 3.5 | 12.5 ± 3.4 | 5.3 ± 2.6 | 7.1 | 7.5 ± 3.7 | 2.8 ± 2.3 | 4.7 |
| Gluteus minimus | 7.6 ± 2.5 | 3.7 ± 2.2 | 3.9 | 4.6 ± 1.5 | 1.6 ± 0.8 | 3.0 | 9.6 ± 2.7 | 4.5 ± 1.9 | 5.1 | 6.4 ± 3.2 | 2.6 ± 1.9 | 3.8 |
| Rectus femoris | 6.4 ± 2.4 | 2.3 ± 1.9 | 4.0 | 3.9 ± 0.7 | 1.1 ± 0.4 | 2.8 | 8.9 ± 2.3 | 3.1 ± 1.2 | 5.7 | 5.0 ± 1.7 | 1.5 ± 0.7 | 3.6 |
| Vastus intermedius | 4.9 ± 1.4 | 2.8 ± 1.3 | 2.1 | 3.0 ± 0.7 | 1.5 ± 0.4 | 1.5 | 6.2 ± 1.8 | 3.4 ± 1.3 | 2.8 | 3.7 ± 1.3 | 2.1 ± 1.0 | 1.6 |
| Vastus medialis | 6.6 ± 1.6 | 3.1 ± 1.1 | 3.5 | 4.4 ± 0.9 | 1.6 ± 0.3 | 2.8 | 7.3 ± 1.7 | 3.5 ± 1.5 | 3.9 | 5.4 ± 1.6 | 2.1 ± 0.8 | 3.3 |
| Vastus lateralis | 5.8 ± 1.4 | 2.2 ± 1.0 | 3.6 | 3.9 ± 0.7 | 1.4 ± 0.4 | 2.5 | 7.4 ± 2.0 | 3.2 ± 1.4 | 4.2 | 4.9 ± 1.5 | 1.8 ± 0.9 | 3.1 |
| Biceps femoris, caput breve | 5.8 ± 1.5 | 3.5 ± 1.8 | 2.2 | 4.1 ± 1.0 | 2.9 ± 1.2 | 1.2 | 8.4 ± 1.5 | 7.2 ± 2.4 | 1.2 | 5.7 ± 1.6 | 3.9 ± 1.7 | 1.8 |
| Biceps femoris, caput longus | 6.3 ± 1.3 | 3.1 ± 1.3 | 3.2 | 4.1 ± 1.2 | 1.8 ± 0.8 | 2.3 | 9.2 ± 2.4 | 5.5 ± 2.3 | 3.7 | 6.0 ± 2.1 | 3.1 ± 1.6 | 2.9 |
| Semimembranosus | 9.2 ± 4.6 | 6.9 ± 5.2 | 2.3 | 4.8 ± 1.1 | 2.4 ± 0.6 | 2.4 | 12.3 ± 3.7 | 9.1 ± 3.7 | 3.2 | 6.8 ± 2.3 | 4.0 ± 2.1 | 2.8 |
| Semitendinosus | 6.4 ± 4.1 | 4.2 ± 3.9 | 2.3 | 3.7 ± 1.2 | 1.7 ± 0.7 | 1.9 | 9.2 ± 2.2 | 6.5 ± 2.4 | 2.7 | 5.6 ± 2.3 | 3.5 ± 2.0 | 2.2 |
| Gracilis | 5.6 ± 1.3 | 2.2 ± 0.9 | 3.3 | 4.2 ± 1.0 | 1.8 ± 0.7 | 2.4 | 8.3 ± 2.0 | 5.1 ± 2.1 | 3.1 | 6.0 ± 2.1 | 3.2 ± 1.6 | 2.8 |
| Sartorius | 6.8 ± 1.6 | 3.8 ± 1.3 | 3.0 | 5.1 ± 0.9 | 2.6 ± 0.8 | 2.5 | 8.5 ± 2.0 | 6.3 ± 2.7 | 2.2 | 6.7 ± 2.5 | 4.1 ± 2.2 | 2.6 |
| Gastrocnemius lateralis | 5.4 ± 1.3 | 3.6 ± 1.4 | 1.8 | 4.2 ± 0.9 | 1.8 ± 0.5 | 2.4 | 8.0 ± 2.4 | 4.9 ± 1.6 | 3.1 | 5.3 ± 1.2 | 2.7 ± 1.0 | 2.6 |
| Gastrocnemius medialis | 5.5 ± 1.7 | 3.5 ± 1.6 | 2.0 | 3.5 ± 0.4 | 2.0 ± 0.4 | 1.6 | 8.3 ± 5.2 | 6.2 ± 3.1 | 2.2 | 4.7 ± 1.1 | 3.1 ± 1.2 | 1.7 |
| Soleus | 6.7 ± 1.9 | 5.3 ± 2.0 | 1.4 | 3.9 ± 0.5 | 2.5 ± 0.5 | 1.4 | 8.8 ± 2.9 | 7.1 ± 2.0 | 1.7 | 5.0 ± 1.4 | 3.4 ± 1.3 | 1.6 |
| Tibialis anterior | 5.6 ± 0.9 | 2.9 ± 1.0 | 2.7 | 3.5 ± 0.5 | 1.4 ± 0.4 | 2.0 | 6.6 ± 2.2 | 3.8 ± 1.4 | 2.8 | 4.5 ± 1.0 | 2.2 ± 0.8 | 2.3 |

The muscle matrices were split into a central (A) and a peripheral shell region (B). The periphery contained the upper and lower slices of a muscle matrix and an outer ring of two voxels (2.34 mm thickness) from the cross-sectional muscle slices in between. The centre consisted of the remaining voxels. For *p*-values and effect sizes for the comparisons between periphery and centre and across the four subject groups, see Table 5.

**Table 5. p-Values resulting from linear mixed effect models testing regional distribution of the FF within each muscle**

| | | | Main effects | | | Univariate tests | | | | | | | |
| | | | | | | Region in group | | | | | | | |
| | | | Region | Group | Region*group | Old athlete | | Old control | | Young athlete | | Young control | |
| Muscle | n | df | p | p | p | p | d | p | d | p | d | p | d |
|---|---|---|---|---|---|---|---|---|---|---|---|---|---|
| Gluteus maximus | 43 | 78 | <0.001 | <0.001 | 0.995 | <0.001 | 0.791 | <0.001 | 0.834 | <0.001 | 0.820 | <0.001 | 0.919 |
| Gluteus medius | 43 | 78 | <0.001 | <0.001 | 0.995 | <0.001 | 0.705 | <0.001 | 0.694 | <0.001 | 0.702 | <0.001 | 0.776 |
| Gluteus minimus | 43 | 78 | <0.001 | <0.001 | 0.984 | <0.001 | 0.618 | <0.001 | 0.729 | <0.001 | 0.636 | <0.001 | 0.730 |
| Rectus femoris | 43 | 78 | <0.001 | <0.001 | 0.668 | <0.001 | 0.790 | <0.001 | 0.863 | <0.001 | 0.594 | <0.001 | 0.725 |
| Vastus intermedius | 43 | 78 | <0.001 | <0.001 | 0.276 | <0.001 | 0.604 | <0.001 | 0.812 | 0.00905 | 0.408 | 0.00207 | 0.486 |
| Vastus lateralis | 43 | 78 | <0.001 | <0.001 | 0.202 | <0.001 | 0.955 | <0.001 | 1.173 | <0.001 | 0.677 | <0.001 | 0.919 |
| Vastus medialis | 43 | 78 | <0.001 | <0.001 | 0.616 | <0.001 | 0.954 | <0.001 | 1.098 | <0.001 | 0.770 | <0.001 | 0.982 |
| Biceps femoris, caput breve | 43 | 78 | <0.001 | <0.001 | 0.710 | 0.00316 | 0.465 | 0.0880 | 0.263 | 0.0975 | 0.256 | 0.00963 | 0.405 |
| Biceps femoris, caput longus[b] | 42 | 76 | <0.001 | <0.001 | 0.631 | <0.001 | 0.599 | <0.001 | 0.759 | 0.00460 | 0.451 | <0.001 | 0.620 |
| Semimembranosus | 43 | 78 | <0.001 | <0.001 | 0.794 | 0.0490 | 0.305 | 0.0497 | 0.304 | 0.00372 | 0.456 | 0.00243 | 0.478 |
| Semitendinosus | 42 | 76 | <0.001 | <0.001 | 0.821 | 0.0587 | 0.296 | 0.102 | 0.255 | 0.00544 | 0.441 | 0.0193 | 0.369 |
| Gracilis | 43 | 78 | <0.001 | <0.001 | 0.770 | <0.001 | 0.720 | <0.001 | 0.707 | 0.00132 | 0.508 | <0.001 | 0.654 |
| Sartorius | 43 | 78 | <0.001 | <0.001 | 0.922 | <0.001 | 0.537 | 0.00873 | 0.410 | 0.00413 | 0.451 | 0.00138 | 0.506 |
| Gastrocnemius lateralis | 43 | 78 | <0.001 | <0.001 | 0.515 | 0.00486 | 0.442 | <0.001 | 0.798 | <0.001 | 0.596 | <0.001 | 0.684 |
| Gastrocnemius medialis | 42 | 76 | <0.001 | <0.001 | 0.920 | <0.001 | 0.563 | 0.0129 | 0.393 | 0.00478 | 0.449 | 0.00123 | 0.519 |
| Soleus | 43 | 78 | <0.001 | <0.001 | 0.984 | 0.0772 | 0.273 | 0.0214 | 0.358 | 0.0778 | 0.273 | 0.0264 | 0.345 |
| Tibialis anterior | 43 | 78 | <0.001 | <0.001 | 0.668 | <0.001 | 0.790 | <0.001 | 0.863 | <0.001 | 0.594 | <0.001 | 0.725 |

Fixed main effects are regions referring to the periphery and the centre region selected for each muscle, group and region*group. Univariate retests used the Bonferroni correction for *p*-value. The significance level was *p* < 0.050; significant values are in red.

Abbreviations: *d*, effect sizes for univariate tests are given as Cohen's *d*; df, degrees of freedom; FF, fat fraction; *n*, number of subjects.

Univariate testing showed that age significantly affects controls (df = 39, $p < 0.001$, $d = 0.812$) but not athletes (df = 39, $p = 0.0702$, $d = 0.284$). Old athletes had lower EMCL than old controls (df = 39, $p < 0.001$, $d = 0.577$), whereas sport had no effects on the EMCL of young muscles (df = 39, $p = 0.543$, $d = 0.094$).

IMCL exhibited significant main effects of age (df = 39, $p = 0.00390$) but not of sport (df = 39, $p = 0.0621$, Fig. 4). Univariate testing confirmed a significant small effect of age in controls (df = 39, $p = 0.0156$, $d = 0.386$) but not in athlete muscles (df = 39, $p = 0.0739$, $d = 0.280$).

In addition to $^1$H-MRS the neutral lipid concentration of the myofibres was determined using oil red O-stained cross-sections of the muscle biopsy specimens. Of the 43 subjects 30 samples contained small-sized, 11 medium-sized and 2 coarse-sized droplets, and the size of the lipid droplets was not different between the four groups ($p = 0.630$; the mean value ± SD of categories was $1.3 ± 0.6$). In conclusion athletic sport reduced the fraction of fat in adipocytes but not in the muscle cells in both age groups.

### Correlation between muscle volume and jumping power

In both age groups the summed volume of the jumping muscles (glutei, quadriceps and triceps surae muscles) correlated positively with the peak power determined during a countermovement jump. However the correlation functions of old and young subjects were significantly different ($p = 0.002$). The correlation between power and volume was weaker in old subjects ($R = 0.67$) than in young subjects ($R = 0.94$), and the increase in power with increasing volume was almost twice as high in the young subjects than in the old subjects (0.89 *vs.* 0.45 W/ml, Table 6; Fig. 5*A*). A study of only the estimated lean muscle volume (Table 6; Fig. 5*B*) showed that the data were naturally shifted towards the left, especially for the older subject groups. However the values of the slope and correlation coefficients were virtually unchanged (Table 6), suggesting that reduction in muscle volume by adipose tissue does not explain the difference in power development between old and young subjects.

## Discussion

Masters athletes have significantly larger volumes in most thigh muscles and in the gluteus maximus muscle than the old control subjects. In contrast no difference in muscle volume was found between the athletes and controls in the lower legs of the old test subjects. When Masters athletes were compared with young athletes for muscle volume, a significantly lower volume of the quadriceps muscle was found in the Masters athletes. A recently published study on old and young people similar to our control subjects showed that in the old subjects a lower muscle volume was found mainly in the thigh muscles, less volume in the hip muscles and nothing in the lower leg muscles (Fuchs et al., 2023). In our study this finding was reproduced as a trend only in the control subjects, which could presumably be due to the smaller number of cases in relation to the controls. However if the controls and athletes are combined in our study, the earlier finding (Fuchs et al., 2023) that the thigh muscles in particular are affected by lower volumes with age is confirmed.

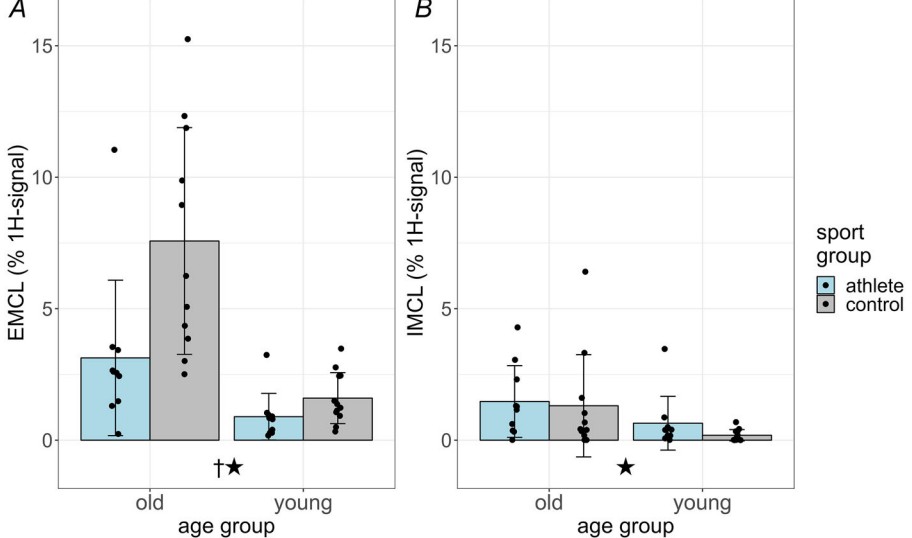

**Figure 4. Lipids in soleus muscle determined by 1H-MR spectroscopy**
Relative concentrations of extramyocellular lipids (EMCLs, *A*) and intramyocellular lipid (IMCL, *B*, % of the total $^1$H-signal, mean ± SD, *n* = 43, all male) in the soleus muscle measured using single-voxel 1H-MR spectroscopy.

**Table 6. Correlations between the peak power (kW) determined by countermovement jumps and the summed volumes (ml) or summed lean volumes (ml) of the jumping muscles (glutei, quadriceps and triceps surae muscles)**

| Power *versus* volume | Intercept (kW) | Slope (W/ml) | R |
|---|---|---|---|
| Old subjects, n = 21 | 0.67 | 0.45 | 0.67 |
| Young subjects, n = 22 | −0.61 | 0.89 | 0.94 |
| Young–old | | 0.45 | |

| Power *versus* lean volume | Intercept (kW) | Slope (W/ml) | R |
|---|---|---|---|
| Old subjects, n = 21 | 0.75 | 0.46 | 0.68 |
| Young subjects, n = 22 | −0.40 | 0.88 | 0.94 |
| Young–old | | 0.42 | |

The lean volume was estimated as the product of volume and water fraction (100 – fat fraction). The *p*-value of all four correlations was <0.001. See also Fig. 5.

The fat fraction of the muscles of the Masters athletes was significantly higher than that of the young athletes. In both age groups competitive sport also led to lower values of fat fraction compared to the controls. The Masters athletes exhibited a fat fraction in all muscles that approximately corresponded to the fat fraction of the young controls.

Our spectroscopic investigations of the soleus muscle showed that the increase in fat fraction in old age occurs mainly in the extramyocellular fat (EMCL), irrespective of sporting activity. In the controls IMCL also increased significantly with age. However IMCL remained below 2% [1]H-signal even with age (Fig. 4). The increase in fat fraction with age is therefore not due to an increase in fat in the muscle fibres but due to the presence of adipocytes. In Masters athletes the EMCL concentration is only half as high as in the older controls.

Changes in EMCL were often not found, or they were so small that they were not detectable in addition to the large fluctuations in the EMCL signal depending on the placement of the measurement voxel in the muscle (Boesch & Kreis, 2000). Increases in muscle fat during the course of acute atrophy, for example, after an injury, were previously also attributed to intramyocellular fat and explained by a reduction in mitochondrial energy metabolism (Flück et al., 2020). However in contrast to these short-term effects influencing muscle metabolism and IMCL, our results suggest that the long-term process of increased myosteatosis due to muscle ageing is primarily due to an increased volume of adipocytes within the muscles. As our results also show the latter was reduced by long-term intensive training.

The histological measurement of the fat droplets in the soleus muscle fibres also underlined the fact that the group differences in the fat fraction originate essentially from the EMCL and not from the IMCL. The size of the oil red O-stained neutral lipids in the muscle fibres showed no group differences, basically corresponding to the analogous results of the IMCL from the spectroscopy in the soleus, which showed only an age difference in the controls. It is possible that [1]H-MRS in muscle is even more sensitive for measuring small differences in the fat content of muscle fibres than histology (Boesch & Kreis, 2000), but this could not be shown directly here, as two different muscles were examined using the two methods and the histological result was obtained from a comparatively small area of tissue.

It has previously been observed that the EMCL signal becomes larger when the measurement voxel includes peripheral areas of the muscle and that the EMCL signal has the lowest amplitude when the voxel is located in the centre of the muscle belly (Boesch & Kreis, 2000). We therefore also placed the voxels for spectroscopy in the centre of the muscle belly of the soleus muscle wherever possible. In the images obtained using MRI, which show the fat fraction as a grey value, no zonation of the fat fraction was visible in the muscle cross-sections except for the gluteus maximus muscle. However a separate analysis of a peripheral zone of the muscle and the central part revealed a significantly higher fat fraction in the periphery than in the centre for almost all muscles and in almost all groups. Notably in the soleus muscle, which we also analysed spectroscopically, this zonation was only minimal and not significant in some groups. In the vastus lateralis muscle the zonation was significant in all test groups, which emphasizes the dependence of the EMCL signal on the positioning of the measuring voxel in this muscle (Boesch & Kreis, 2000).

The fat fraction of a muscle is not determined only by sport and age. The anatomical position of the muscle appears to play a significant role in controlling the fat fraction of a muscle. In all four test groups the more dorsal and proximal the muscles are, the higher their fat fraction. The gluteus maximus muscle had the highest fat fraction in all four groups. In the thigh we confirm earlier findings that the knee flexors have a higher fat

fraction than the knee extensors (Grimm et al., 2018a), whereby we showed that this fundamental, anatomical difference in the fat fraction occurs independently of age and competitive sport. A comparison of the gluteus maximus muscle with the quadriceps muscle showed how strongly the anatomical position alone determines the fat fraction. Both muscles carry out a heavy workload during locomotion. However the gluteus maximus is the muscle with the highest fat fraction, and the four heads of the quadriceps belong to the muscles with a very low fat fraction. A comparison of the gluteus muscles with each other also clearly showed the significance of the pure anatomical position on the fat fraction, as the gluteus medius and the gluteus minimus are also lean muscles.

The significantly reduced peak power during a counter-movement jump in the older subjects compared to the young subjects confirms earlier findings that the maximum performance development of the muscles was significantly reduced in the older people compared to the young people (Alvero-Cruz et al., 2021; Ireland et al., 2022;

Runge et al., 2004). The fact that the correlation between muscle volume and peak power in the Masters athletes was the same as that in the older controls suggests that the reduced power development of the muscles in the Masters athletes is subject to the same problems as in the old controls. The Masters athletes predominantly achieved a higher jumping power than the old controls due to a larger muscle volume. Our results further show that the age-related difference in jumping performance cannot be explained only by the reduction in muscle volume due to the increased fat content in the muscle, at least at the level of adiposity measured in the Masters athletes and the older controls.

In addition to age-related myosteatosis, our results contain evidence of age-related sarcosthenia, that is, reduction in intrinsic power-generating potential, as evidenced by the clearly distinguished regression lines in old and young subjects (Fig. 5). The fact that accounting for myosteatosis left the relationship between muscle mass and peak power virtually unchanged demonstrates that

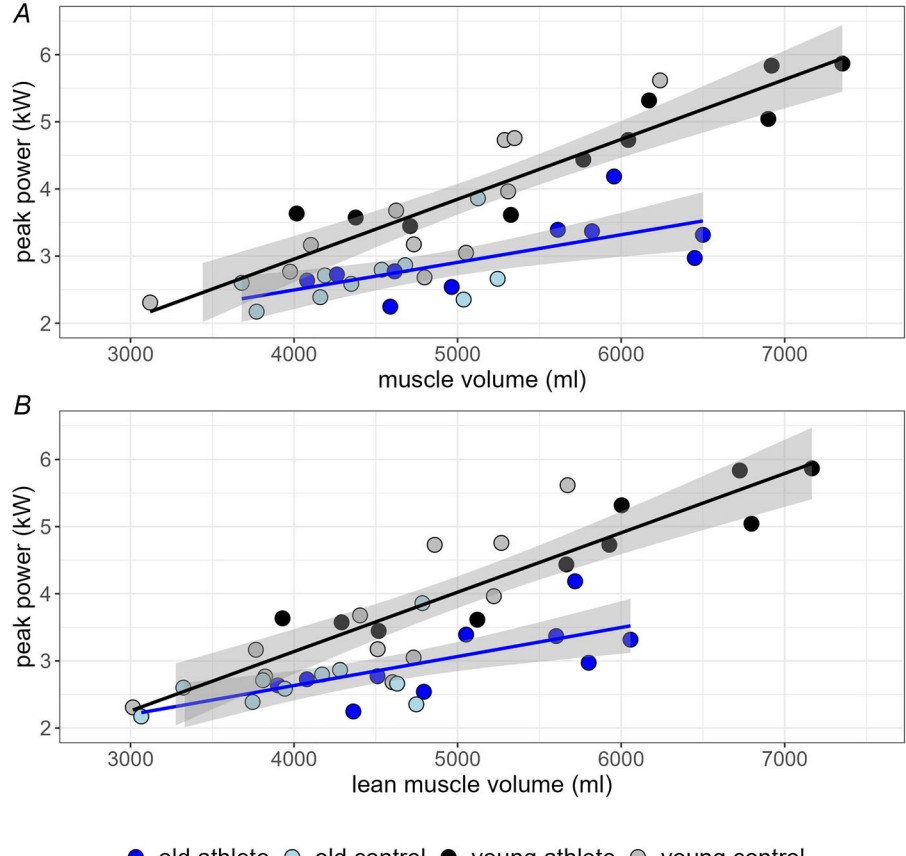

**Figure 5. Correlation between summed muscle volume (*A*, ml) and summed lean muscle volume (*B*, ml) of the glutei, quadriceps and triceps surae muscles *versus* the peak power (kW) determined during a countermovement jump**
Lean muscle volume was calculated as the muscle volume times the average water fraction (100 − fat fraction). Old athletes, *n* = 10; old controls, *n* = 11; young athletes, *n* = 10; young controls, *n* = 12. All subjects are male. See also Table 6.

**Table 7. Volume (ml, mean ± SD) and lean volume (ml) of jumping muscles and the peak power (kW) determined at a counter-movement jump**

| Group | Volume (ml) | Lean volume (ml) | Peak power (kW) |
|---|---|---|---|
| Old athletes, *n* = 10 | 5285 ± 895 | 4988 ± 774 | 3.02 ± 0.56 |
| Old controls, *n* = 11 | 4379 ± 612 | 3957 ± 637 | 2.63 ± 0.50 |
| Young athletes, *n* = 10 | 5758 ± 1139 | 5614 ± 1122 | 4.55 ± 0.95 |
| Young controls, *n* = 12 | 4781 ± 833 | 4534 ± 767 | 3.63 ± 1.03 |

The volume contains the glutei, quadriceps and triceps surae muscles. The lean volume is volume times the water fraction (100 – fat fraction). See also Figs 5 and 6.

myosteatosis and sarcosthenia are different entities. To illustrate this with an example, muscle volume differed between young and old athletes by 473 ml (5758 *vs.* 5285 ml; Table 7). Using a slope of 0.89 kW/ml (Table 6), age-related sarcopenia therefore explains a power decline by 421 Watts. At the same time older athletes had 153 ml more intramuscular lipids in their leg muscle, which explains a further power deficit of 136 Watts. Thus sarcopenia and the age-related exaggeration of myosteatosis jointly explain 557 Watts (36%) of the 1534 Watts in power decline in athletes (Table 7), leaving 64% to be explained by sarcosthenia. For control participants the relative contributions of sarcopenia, myosteatosis and sarcosthenia are very similar to the athletes (Fig. 6).

In this context the traditional view of an age-related decline in type II-fibre size and abundance comes into play (Lexell, 1995). Indeed evidence from muscle biopsy data does demonstrate a decline in type II-fibre area in Masters athletes (Messa et al., 2020). However that decline amounts to ∼25% only between 25 and 70 years of age, whereas regression slopes in Fig. 5 indicate a difference by 50%. This may suggest that factors other than type II-fibre shrinkage may also contribute to the occurrence of sarcosthenia, such as rarefication of contractile proteins or their exaggerated posttranslational modification (D'Antona et al., 2003; Gilliver et al., 2010). The clear and distinctly different regression lines for young and old people in Fig. 5*A*, the left shift of the data in Fig. 5*B* and the fact that gross muscle size seemed more or less comparable between young and older athletes suggest that sarcopenia (i.e. muscle shrinkage *per se*) can potentially be halted by sprint and jump training into older age but not so myosteatosis and sarcosthenia.

The present study has some limitations. Firstly the recruitment aim of 12 participants per group could not be achieved. This was due to the fact that the study was performed during the COVID-19 pandemic, which heavily impacted the availability of participants and access to the laboratory. However we feel that this limitation had no major effect on the outcome, given that good matching of groups was achieved, and that highly significant results were obtained. Second results need to be interpreted *cum grano salis*, as the study design was of cross-sectional nature, and given that participants self-selected themselves to the different groups. However testing our hypothesis in a randomized controlled longitudinal trial would require immense resources and likely not be feasible. We therefore argue that the data presented here provide the best evidence for beneficial effects of prolonged high-level engagement in jump and spring track and field events to offset sarcopenia and myosteatosis.

We thus conclude that male Masters athletes specializing in sprint and jump events are largely able to compensate for age-related deficits in leg and hip muscle volume, probably through their intensive training regimen. Furthermore Masters athletes partially mitigate the occurrence of myosteatosis but not the age-related sarcosthenia. Age-related myosteatosis is

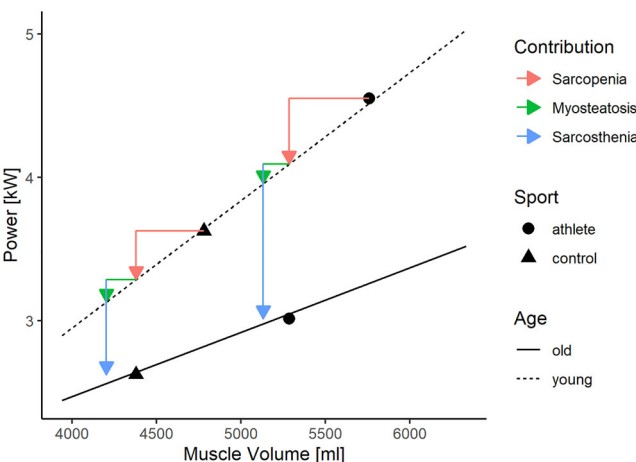

**Figure 6. Projection of regression parameters onto contributions to dynapenia (i.e. age-related power deficits) by sarcopenia, myosteatosis and sarcosthenia, for both athletes and control participants (black symbols are group means)**
Sarcopenia and myosteatosis both denote decrements in effective muscle volumes, which were transformed into power decrements using the regression slope of 0.89 W/ml from Table 7. The remaining gap in the power deficit denotes sarcosthenia, that is, the volume-invariant power deficit.

mostly an effect of extramyocellular adipocytes and not of IMCL stores. The accumulated fat will certainly increase inertia of the body during movements and thus limit the neuromuscular power output (Alvero-Cruz et al., 2021; Sanchez-Trigo et al., 2022) and contribute to dynapenia. However age-related sarcopenia and sarcosthenia occurred in athletes and control participants to a comparable extent. Quantitatively sarcosthenia was the greatest contributor to dynapenia. The exact causes of sarcosthenia remain to be determined. Similarly the mechanistic links between myosteatosis and sarcopenia remain to be elucidated.

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

## Additional information

### Data availability statement

Individual data collected for this study are presented in the figures and tables. Access to additional data will be available from the corresponding author upon reasonable request.

### Competing interests

The authors declare that they have no competing interests.

### Author contributions

Conceptualization: J.Z. and J.R.; acquisition and analysis of data: J.Z., J.E. and C.S.C; interpretation of data: J.Z., C.S.C. and J.R.; drafting and revising the manuscript: J.Z., C.S.C. and J.R. All authors have read and agreed to the published version of the manuscript, and all persons qualified for authorship are listed.

### Funding

This research did not receive external funding.

### Acknowledgements

The authors thank Carolin Berwanger, Institute of Aerospace Medicine, German Aerospace Centre, Cologne, Germany, for histological staining of the muscle cryosections, and Rolf Schröder, Institute of Neuropathology, University Hospital Erlangen, Friedrich-Alexander University Erlangen-Nürnberg, Erlangen, Germany, for their myopathological evaluation. Last but not least the authors thank the study participants. Without their selfless contribution this work would not have been possible.

### Keywords

aging, magnetic resonance imaging, masters athletes, muscle physiology, myosteatosis, sarcopenia, skeletal muscle

### Supporting information

Additional supporting information can be found online in the Supporting Information section at the end of the HTML view of the article. Supporting information files available:

**Peer Review History**

