## [Peer Review History · The Journal of Physiology]

Leg and hip muscles show muscle specific effects of ageing and sport on muscle volume and fat fraction in Masters athletes

Jochen Zange, Joachim Endres, Christoph Stephan Clemen, and Jörn Rittweger
DOI: 10.1113/JP285665

Corresponding author(s): Jochen Zange (Jochen.Zange@dlr.de)

Review Timeline:

Submission Date:	18-Jun-2024
Editorial Decision:	04-Sep-2024
Revision Received:	28-Nov-2024
Editorial Decision:	20-Jan-2025
Revision Received:	19-Feb-2025
Accepted:	03-Mar-2025

Senior Editor: Karyn Hamilton

Reviewing Editor: Christoph Centner

Transaction Report:

Dear Dr Zange,

Re: JP-RP-2024-285665 "Leg and hip muscles show muscle specific effects of ageing and sport on muscle volume and fat fraction in Masters athletes" by Jochen Zange, Joachim Endres, Christoph Stephan Clemen, and Jörn Rittweger

Thank you for submitting your manuscript to The Journal of Physiology. It has been assessed by a Reviewing Editor and by 2 expert referees and we are pleased to tell you that it is potentially acceptable for publication following satisfactory major revision.

REVISION CHECKLIST:

Please upload two versions of your manuscript text: one with all relevant changes highlighted and one clean version with no

changes tracked. The manuscript file should include all tables and figure legends, but each figure/graph should be uploaded as separate, high-resolution files.

We look forward to receiving your revised submission.

Yours sincerely,

Karyn Hamilton
Senior Editor
The Journal of Physiology

REQUIRED ITEMS

- Author photo and profile. First or joint first authors are asked to provide a short biography (no more than 100 words for one author or 150 words in total for joint first authors) and a portrait photograph. These should be uploaded and clearly labelled together in a Word document with the revised version of the manuscript. See Information for Authors for further details.

- You must start the Methods section with a paragraph headed Ethical Approval. If experiments were conducted on humans, confirmation that informed consent was obtained, preferably in writing, that the studies conformed to the standards set by the latest revision of the Declaration of Helsinki and that the procedures were approved by a properly constituted ethics committee, which should be named, must be included in the article file. If the research study was registered (clause 35 of the Declaration of Helsinki), the registration database should be indicated, otherwise the lack of registration should be noted as an exception (e.g. The study conformed to the standards set by the Declaration of Helsinki, except for registration in a database). For further information see: <https://physoc.onlinelibrary.wiley.com/hub/human-experiments>.

- Your manuscript must include a complete Additional Information section, including competing interests; funding; author contributions and acknowledgements.

- Please ensure that any tables are editable and in Word format, and wherever possible, embedded in the article file itself.

- Please ensure that the Article File you upload is a Word file.

- Papers must comply with the Statistics Policy: https://jp.msubmit.net/cgi-bin/main.plex?form_type=display_requirements#statistics.

In summary:

- If $n \leq 30$, all data points must be plotted in the figure in a way that reveals their range and distribution. A bar graph with data points overlaid, a box and whisker plot or a violin plot (preferably with data points included) are acceptable formats.

- If $n > 30$, then the entire raw dataset must be made available either as supporting information, or hosted on a not-for-profit repository, e.g. FigShare, with access details provided in the manuscript.

- 'n' clearly defined (e.g. x cells from y slices in z animals) in the Methods. Authors should be mindful of pseudoreplication.

- All relevant 'n' values must be clearly stated in the main text, figures and tables.

- The most appropriate summary statistic (e.g. mean or median and standard deviation) must be used. Standard Error of the Mean (SEM) alone is not permitted.

- Exact p values must be stated. Authors must not use 'greater than' or 'less than'. Exact p values must be stated to three significant figures even when 'no statistical significance' is claimed.

- A Data Availability Statement is required for all papers reporting original data. This must be in the Additional Information section of the manuscript itself. It must have the paragraph heading 'Data Availability Statement'. All data supporting the results in the paper must be either: in the paper itself; uploaded as Supporting Information for Online Publication; or archived in an appropriate public repository. The statement needs to describe the availability or the absence of shared data. Authors must include in their statement: a link to the repository they have used, or a statement that it is available as Supporting Information; reference the data in the appropriate sections(s) of their manuscript; and cite the data they have shared in the References section. Whenever possible, the scripts and other artefacts used to generate the analyses presented in the paper should also be publicly archived. If sharing data compromises ethical standards or legal requirements then authors are not expected to share it, but must note this in their statement. For more information, see our Statistics Policy.

- Please include an Abstract Figure file, as well as the Figure Legend text within the main article file. The Abstract Figure is a piece of artwork designed to give readers an immediate understanding of the research and should summarise the main conclusions. If possible, the image should be easily 'readable' from left to right or top to bottom. It should show the physiological relevance of the manuscript so readers can assess the importance and content of its findings. Abstract Figures should not merely recapitulate other figures in the manuscript. Please try to keep the diagram as simple as possible and without superfluous information that may distract from the main conclusion(s). Abstract Figures must be provided by authors no later than the revised manuscript stage and should be uploaded as a separate file during online submission labelled as File Type 'Abstract Figure'. Please also ensure that you include the figure legend in the main article file. All Abstract Figures should be created using BioRender. Authors should use The Journal's premium BioRender account to export high-resolution images. Details on how to use and access the premium account are included as part of this email.

EDITOR COMMENTS

Reviewing Editor:

We thank the authors for submitting their interesting study. The points raised by the reviewers emphasize the need to better align findings with current literature, refine the clarity and detail in the abstract and introduction, and address methodological concerns, such as participant health status, training history, and measurement accuracy. Furthermore, I recommend that the discussion should more thoroughly connect the study's results to existing research, especially regarding muscle volume differences across groups.

Please also see 'Required Items' above.

Ethics Concerns:

No information about informed consent is provided.

Senior Editor:

Thank you for submitting your manuscript for consideration by The Journal of Physiology for a Special Issue. As part of the peer review process, we recruited two Referees with expertise in this field of study. As you can see, they've each provided very detailed feedback. We would like to give you the opportunity to address each of the points raised by the Referees, making corresponding revisions to your manuscript. Please make certain to include a clear statement indicating the process for written informed consent for these human participants in your revised manuscript. In addition, we would like to have you revisit the Statistics Policy to verify that your revised manuscript is fully in line with The Journal's policies and provides the clearest statistics reporting possible. Thank you for your interest in The Journal and we look forward to seeing your revised manuscript.

Comments for Authors to ensure the paper complies with the Statistics Policy (Required):

We appreciate that the authors seem to be trying to include precise p-values as required by the Journal's Statistics Policy. However, it seems a bit confusing to use notation like $0.000 < P < 0.0458$ or 0.0095

REFeree COMMENTS

Referee #1:

This is an interesting study of skeletal muscle volumes and fat fractions in young and older masters athletes and inactive counterparts. It is well written overall, well presented and the methodology and statistical analysis are appropriate. The research question is important and the findings will be influential and have impact in the research field. I only have a few relatively minor comments:

Some of the figures in the main manuscript were hazy and should be improved, as they are difficult to distinguish or to interpret. E.g. Fig 3. Although, I noted the improved quality Figs at the end of the manuscript, so it seems this is in hand.

Start of the Discussion section does not fully reflect the findings. The authors suggest that "Masters athletes are able, at least to some extent, to maintain their muscle volume compared to young athletes". ..."However, the fat fraction of the muscles of the Masters athletes was significantly higher than the fat fraction of the young athletes". Readers may take issue with these statements. From the results, it can be seen that being involved in competitive sports was associated with larger muscle volumes in young and older age compared with the age-matched athletic or inactive counterparts (as per the effect of Sport in Table 2). There was also an effect of age for 8 muscles examined, which Figure 1 shows as older subjects having lower volumes than young overall (and Table 6 shows as being about 8 - 10%). There was no age x sport interaction for any of the muscles examined. Therefore, doesn't this mean that age has a similar effect on these muscles, regardless of whether the subject is athletic or not? If this interpretation is correct, it should not be explicitly stated in the opening sentence of the discussion that masters athletes are able to maintain their muscle volume compared to young athletes.

There is a similar point to be made towards the end of the opening paragraph of the discussion. The authors suggest that competitive sports led to a reduction in fat fraction (this is not strictly correct, since a reduction has not been measured. Instead, a difference between groups has been measured at a single time point). Also that masters athletes had a fat fraction in all muscles roughly corresponding with the fat fraction of young controls. However, they miss the point that masters athletes had higher fat fraction compared with young athletes. For completeness, and to avoid a possible bias of the narrative, this point needs to be made. From a different perspective, the findings show that masters athletes have lower muscle volumes and higher fat fractions than younger athletes, so they clearly succumb to the typical effects of ageing, albeit their exercise confers clear benefits for muscle and fat compared with those not engaged in sports.

Some minor amendments to the discussion text would avoid these issues with the narrative.

Referee #2:

The present study examines differences in muscle volume, muscle fat fraction, and jump performance between older and younger individuals, as well as between trained and untrained groups. The presented data is interesting and contributes to the existing literature. However, it is important to better align the findings with the current literature to clearly demonstrate their novelty and contribution to current knowledge.

Please refer to the attached comments for detailed suggestions. I also have questions and suggestions regarding several aspects of the manuscript that need to be addressed.

General comment: as you did not assess females, include 'male' in your title and where appropriate.

Abstract

Include the numbers per group as well as some characteristics.

I would advise changing the second sentence to make it clearer, for example: "...athletes prevent the age-related loss in muscle volume and performance as well as the increase in muscle fat ..."

I recommend stating that MRI was used (instead of "...using a 6-point DIXON sequence."), as the readership of the Journal of Physiology will not all be familiar with specific MR sequences. MRI will be more appropriate in the abstract. Using both is also fine: "...using Magnetic Resonance Imaging (MRI) with a 6-point...."

For clarity, add briefly (can be between brackets) what 'lean volume' is.

Please add more data in the abstract, not only P values. For example, what is the muscle volume in the different groups? For the quadriceps? Hamstrings? Make sure that the reader already gets some more information based on the abstract.

Consider adding a clear(er) conclusion in the abstract.

Introduction

"Long-term inactivity, e.g. by an....magnetic resonance imaging." You can make this statement even stronger, as shorter periods of bed rest have already shown these effects (PMID: 38411283). This demonstrates that relatively brief periods of inactivity, which may be more relevant to aging, can already cause an increase in muscle fat infiltration.

Paragraph 2: "Masters athletes.... non-athletes.."

Some relevant (recent) Masters athlete literature is missing for this section:

PMID: 29715523; 38458181

"However, it is not known..... hips and the legs." This statement is inaccurate. Data comparing different muscle groups (within legs and hip) between young and older individuals exist (Fuchs et al., 2023, PMID: 36967049). Including this will make your narrative more complete. Your study expands on this work by not only comparing young vs. older individuals but also young and older 'athletes'.

It is also not entirely clear from the introduction why 'Masters athletes in power disciplines' were specifically chosen. Clarifying this point would strengthen your introduction.

Methods

Specify in the methods section that 43 participants were included out of the planned 48 due to COVID-19, so this is immediately clear to the readers.

Out of curiosity, (looking at the trial register) you planned to investigate other measurements with the biopsy (collagen) and other measurements (ultrasound). Is there a specific reason this data is not included in this manuscript?

According to the trial register, you planned to include runners in the athletic groups (inclusion criteria: participation in running competitions and regular running training). However, in the manuscript, you focus on power athletes (jumpers, sprinters, etc.). Why did this change?

Do the authors have any insight into the health status of the participants? Were they all considered healthy (specifically the older controls)? No T2DM etc.?

Can you provide more information on the training history and characteristics of the participants (e.g. there was no measure of fat%)? How many years of training? How many times a week? What type of training? What type of athlete? Competitive level, etc.?

Given the different body sizes (Table 1) and your MR settings (260 axial images of 0.5 cm), were you able to scan the entire volume of all pelvic (Glutei) muscles for all individuals?

Do you have any insights into the accuracy of this semi-automatic measurement for assessing muscle volume? Please provide more details. Also, how long did the analysis take to quantify muscle volumes for this study?

You forgot to mention the hamstring muscle volumes in your methods section.

"Moreover, this selection.....joint stabilizers." It would be helpful for readers if you could specify which muscles you are referring to for locomotion and stabilizers (e.g., in brackets).

"We excluded three muscles.....report any symptoms." Do you have any explanation for why those muscles showed such a large fat%?

"The integrals of the water.....see Figure 3) as described....". Did the authors mean to refer to Figure 4?

Were biopsies taken directly after the MRS measurement? Was there a specific reason why MRS was performed on the right calf and biopsies were taken from the left calf?

More information needs to be provided in Paragraph 4.4 to explain your analysis procedure in more detail.

Results

Ensure consistency in your decimal values (e.g., use 3 decimals throughout). Also, correct values such as 'P=844' to a standard format (e.g., P=0.844).

Where appropriate, mention percentage differences in the results section to give readers insight into the magnitude of differences. For example, indicate the percentage impact of age on the quadriceps and hamstrings.

As stated in the legend of Figure 6, specify in the results section which muscles were included as 'jumping muscles'.

Discussion

The discussion largely focuses on muscle fat fraction, which is indeed an important aspect of the present study. However, it would benefit from a greater emphasis on how the current findings (both in muscle volume and fat fraction) align with existing knowledge.

A major and important aspect of the study is the differences in muscle volume between groups (and is also in the title of the manuscript). There is only limited discussion of these findings. For example, how does the data from the present study align with the literature (e.g., Fuchs et al. 2023; PMID: 36967049)? Compared to that study, sex, age, height, body mass, etc., are quite comparable. Bringing your findings in light of current knowledge is important. Your study suggests that thigh muscles are more affected by age compared to other muscle groups, which seems consistent with previous work (Fuchs et al., 2023). Focus more on these aspects, as they form an important take-home message.

For example, irrespective of being active or not, thigh muscles seem to be the most impacted by age compared to other muscle groups. Of course, it needs to be acknowledged that there are limitations with a low sample size and the lack of detailed knowledge on the exact training regimen of all athletes (or please add more information on this in the methods, if available). Nevertheless, your data provides important further insight.

"...whereby the control subjects....activity." Add more details about this in the methods section. Also, in this sentence, change "do" to "did."

"The Masters athletes.... muscle volume." Although this may be true, it is based on a correlation, so there could be other factors at play that were not measured (e.g., neuromuscular factors). Consider toning down this statement.

"In contrast, earlier....the IMCL...". This does not seem particularly relevant to your storyline. I would suggest focusing the discussion on the main findings and put them into perspective with age-related differences in muscle volume and fat fraction, rather than discussing acute (endurance) effects.

Paragraph 5 ("It has previously....(Boesch & Kreis, 2000)."). What is the explanation why fat fraction is higher in the periphery compared to central part of the muscles? Please include more information on this.

Paragraph 6: Can the authors elaborate more on the findings regarding dorsal vs. proximal muscles? Apart from sitting, are there any other reasons that could explain these differences?

Tables/Figures

Figures 1 & 2: Was there a reason why the authors chose to split the figure into two panels? As a suggestion, would the data be clearer if presented in four panels (A-D)? For example, one panel for hip muscles, two panels for thigh muscles, and one panel for lower leg muscles.

Figure 3: Would this data not be clearer if provided in a table? Consider presenting the data in a table format for better clarity and readability.

END OF COMMENTS

EDITOR COMMENTS

Reviewing Editor:

We thank the authors for submitting their interesting study. The points raised by the reviewers emphasize the need to better align findings with current literature, refine the clarity and detail in the abstract and introduction, and address methodological concerns, such as participant health status, training history, and measurement accuracy. Furthermore, I recommend that the discussion should more thoroughly connect the study's results to existing research, especially regarding muscle volume differences across groups.

Please also see 'Required Items' above.

Ethics Concerns:

No information about informed consent is provided.

The first chapter of the methods has been renamed to “Ethical approval”. The information about informed consent has been added.

Senior Editor:

Thank you for submitting your manuscript for consideration by The Journal of Physiology for a Special Issue. As part of the peer review process, we recruited two Referees with expertise in this field of study. As you can see, they've each provided very detailed feedback. We would like to give you the opportunity to address each of the points raised by the Referees, making corresponding revisions to your manuscript. Please make certain to include a clear statement indicating the process for written informed consent for these human participants in your revised manuscript.

This has been added.

In addition, we would like to have you revisit the Statistics Policy to verify that your revised manuscript is fully in line with The Journal's policies and provides the clearest statistics reporting possible. Thank you for your interest in The Journal and we look forward to seeing your revised manuscript.

Comments for Authors to ensure the paper complies with the Statistics Policy (Required):

We appreciate that the authors seem to be trying to include precise p-values as required by the Journal's Statistics Policy. However, it seems a bit confusing to use notation like $0.000 < P < 0.0458$ or

0.0095<P<0.0480 or 0.000 < P 0.0340. Is there a way to provide precise p-values for each comparison rather than these confusing ranges?

The relevant sentences have been rewritten so that it is now recognisable which muscles are meant as an example. It has also been clearly indicated that all P values can be found in one of the corresponding tables.

In Table 1 could you please confirm in the caption that you are listing mean +/- SD?

This information was shifted to the caption.

in your figures is it possible to include precise p-values rather than just indicating "significant"? Or are all comparisons included in the tables (for example, are all comparisons for Figure 1 included in Table 2?)

Figure 1 and 2 have too many P-values to be included as text in the figure. In figure 1 for clarification the sentence "See also Table 2" has been replaced by "The P-values for the effects of sport and age are listed in Table 2." Reference was made to Table 3 in Figure 2.

It seems that Figure 5 does not have a table with precise p-values reflecting what is only listed in the figure legend as "significant." Please revisit the manuscript to ensure full compliance with the Statistics policy. Thank you so much!

In figure 5, the P-values have been inserted into the figure as a number. The values were previously only mentioned in the text. A separate table therefore does not seem necessary to us.

REFEREE COMMENTS

Referee #1:

This is an interesting study of skeletal muscle volumes and fat fractions in young and older masters athletes and inactive counterparts. It is well written overall, well presented and the methodology and statistical analysis are appropriate. The research question is important and the findings will be influential and have impact in the research field. I only have a few relatively minor comments:

Some of the figures in the main manuscript were hazy and should be improved, as they are difficult to distinguish or to interpret. E.g. Fig 3. Although, I noted the improved quality Figs at the end of the

manuscript, so it seems this is in hand.

This seems to be a technical problem with the conversion of the Word text into a pdf document by the publisher's software. In the original Word document, however, the illustrations are sharp.

Start of the Discussion section does not fully reflect the findings. The authors suggest that "Masters athletes are able, at least to some extent, to maintain their muscle volume compared to young athletes". ... "However, the fat fraction of the muscles of the Masters athletes was significantly higher than the fat fraction of the young athletes". Readers may take issue with these statements. From the results, it can be seen that being involved in competitive sports was associated with larger muscle volumes in young and older age compared with the age-matched athletic or inactive counterparts (as per the effect of Sport in Table 2). There was also an effect of age for 8 muscles examined, which Figure 1 shows as older subjects having lower volumes than young overall (and Table 6 shows as being about 8 - 10%). There was no age x sport interaction for any of the muscles examined. Therefore, doesn't this mean that age has a similar effect on these muscles, regardless of whether the subject is athletic or not? If this interpretation is correct, it should not be explicitly stated in the opening sentence of the discussion that masters athletes are able to maintain their muscle volume compared to young athletes.

There is a similar point to be made towards the end of the opening paragraph of the discussion. The authors suggest that competitive sports led to a reduction in fat fraction (this is not strictly correct, since a reduction has not been measured. Instead, a difference between groups has been measured at a single time point). Also that masters athletes had a fat fraction in all muscles roughly corresponding with the fat fraction of young controls. However, they miss the point that masters athletes had higher fat fraction compared with young athletes. For completeness, and to avoid a possible bias of the narrative, this point needs to be made. From a different perspective, the findings show that masters athletes have lower muscle volumes and higher fat fractions than younger athletes, so they clearly succumb to the typical effects of ageing, albeit their exercise confers clear benefits for muscle and fat compared with those not engaged in sports.

Some minor amendments to the discussion text would avoid these issues with the narrative.

The first section of the discussion has also been rewritten considering the comments of the other reviewer.

"Masters athletes have significantly larger volumes in most thigh muscles and in the gluteus maximus muscle than the old control subjects. In contrast, no difference in muscle volume was found between

the athletes and controls in the lower legs of the old test subjects. When comparing Masters athletes with young athletes in terms of muscle volume, a significantly lower volume of the quadriceps muscle was found in the Masters athletes. A recently published study on old and young people similar to our control subjects showed that in the old subjects a lower muscle volume was found mainly in the thigh muscles, less in the hip muscles and not in the lower leg muscles (Fuchs et al., 2023). In our study, this finding was only reproduced as a trend in the control subjects, which could presumably be due to the smaller number of cases in relation to the controls. However, if the controls and athletes are combined in our study, the earlier finding (Fuchs et al., 2023) that the thigh muscles in particular are affected by lower volumes with age is confirmed.

The fat fraction of the muscles of the Masters athletes was significantly higher than the fat fraction of the young athletes. In both age groups, competitive sport also led to lower values of fat fraction compared to the controls. The Masters athletes showed a fat fraction in all muscles that roughly corresponded to the fat fraction of the young controls.”

Referee #2:

The present study examines differences in muscle volume, muscle fat fraction, and jump performance between older and younger individuals, as well as between trained and untrained groups. The presented data is interesting and contributes to the existing literature. However, it is important to better align the findings with the current literature to clearly demonstrate their novelty and contribution to current knowledge.

Please refer to the attached comments for detailed suggestions. I also have questions and suggestions regarding several aspects of the manuscript that need to be addressed.

General comment: as you did not assess females, include 'male' in your title and where appropriate. The word “male” has been added to the title, table 1, figures 1, 2, 3, 5 and 6, and at the beginning and conclusion of the discussion.

Abstract

Include the numbers per group as well as some characteristics.

The numbers have been added and the controls have been characterized as recreationally active.

I would advise changing the second sentence to make it clearer, for example: "...athletes prevent the age-related loss in muscle volume and performance as well as the increase in muscle fat ..."

We have adopted your wording. Thank you very much.

I recommend stating that MRI was used (instead of "...using a 6-point DIXON sequence."), as the readership of the Journal of Physiology will not all be familiar with specific MR sequences. MRI will be more appropriate in the abstract. Using both is also fine: "...using Magnetic Resonance Imaging (MRI) with a 6-point...."

We have chosen your second proposal.

For clarity, add briefly (can be between brackets) what 'lean volume' is.

We have added in brackets: "(volume minus calculated fat volume)"

Please add more data in the abstract, not only P values. For example, what is the muscle volume in the different groups? For the quadriceps? Hamstrings? Make sure that the reader already gets some more information based on the abstract.

The volumes of quadriceps and hamstrings of Masters and young athletes and the range of values of fat fraction has been added.

Consider adding a clear(er) conclusion in the abstract.

The abstract now ends with the following sentences: "In countermovement jumps the muscle peak power per muscle volume was around 50 % lower in old athletes and controls compared to the respective younger comparison groups. This age-related loss of performance can only be explained to a very small extent by reduction in lean portion of the muscle volume due to the increased muscle fat content."

Introduction

"Long-term inactivity, e.g. by an.....magnetic resonance imaging." You can make this statement even stronger, as shorter periods of bed rest have already shown these effects (PMID: 38411283). This demonstrates that relatively brief periods of inactivity, which may be more relevant to aging, can already cause an increase in muscle fat infiltration.

The sentence has been changed as follows: "Long-term inactivity, by 14 days (Fuchs *et al.*, 2024a) or 60 days (De Martino *et al.*, 2021; De Martino *et al.*, 2022) experimental bedrest intervention, increased the fat fraction of skeletal muscles as assessed by magnetic resonance imaging ..."

Paragraph 2: "Masters athletes.... non-athletes.."

Some relevant (recent) Masters athlete literature is missing for this section:

PMID: 29715523; 38458181

The two articles have now also been cited:

“(Hawkins et al., 2003; Narici et al., 2004; Drey et al., 2016; **McKendry et al., 2018**; Pollock et al., 2018; Joanisse et al., 2020).”

“A recent case study of a female Masters athlete who started strength training at the age of 63 showed that old people can not only maintain a high training status into old age, but can also become successful Masters athletes if they only start training in old age (Fuchs et al., 2024b).”

"However, it is not known..... hips and the legs." This statement is inaccurate. Data comparing different muscle groups (within legs and hip) between young and older individuals exist (Fuchs et al., 2023, PMID: 36967049). Including this will make your narrative more complete. Your study expands on this work by not only comparing young vs. older individuals but also young and older 'athletes'. The sentence “However, it is not known ...” has been deleted. The findings by Fuchs et. 2023 on different volume loss in hips, upper and lower legs have been implemented in the first paragraph of the introduction.

It is also not entirely clear from the introduction why 'Masters athletes in power disciplines' were specifically chosen. Clarifying this point would strengthen your introduction.

We added following paragraph:

“The muscles of Masters athletes also adapt to the respective discipline. For example, it has been observed for that jumping performance is better maintained in sprint-trained Masters athletes than in endurance-trained Masters athletes (Michaelis et al., 2008). In addition, Masters athletes who compete in sprint disciplines differ from endurance athletes in terms of their lean mass and the fat content of their bodies (Walker et al., 2023).

In this study, we therefore only selected Masters athletes who participate in strength disciplines in order to analyse a homogeneous group in terms of muscle performance and body composition. It was determined to which extent ...”

Methods

Specify in the methods section that 43 participants were included out of the planned 48 due to COVID-19, so this is immediately clear to the readers.

Following sentence has been added:

“The originally planned 48 participants, i.e. 12 per group, were unfortunately not reached due to the problems caused by the COVID-19 pandemic.”

Out of curiosity, (looking at the trial register) you planned to investigate other measurements with the biopsy (collagen) and other measurements (ultrasound). Is there a specific reason this data is not included in this manuscript?

We are planning another manuscript on the subject of connective tissue, which will also include data from other studies.

According to the trial register, you planned to include runners in the athletic groups (inclusion criteria: participation in running competitions and regular running training). However, in the manuscript, you focus on power athletes (jumpers, sprinters, etc.). Why did this change?

It was not possible to put together a sufficiently large group of pure sprinters for the Masters athletes. This is why we expanded to include jumping disciplines, as the physiology of the athletes is relatively similar here.

Do the authors have any insight into the health status of the participants? Were they all considered healthy (specifically the older controls)? No T2DM etc.?

Following sentence has been added: “Following exclusion criteria were applied: smoking, diabetes mellitus (estimated based on available fasted serum glucose levels), a history of cardiovascular disease or any other condition that could have impact on the musculoskeletal system.”

Can you provide more information on the training history and characteristics of the participants (e.g. there was no measure of fat%?)? How many years of training? How many times a week? What type of training? What type of athlete? Competitive level, etc.?

The last sentence of the chapter "Ethical approval" now describes in more detail what information about the test subjects can be found in the previous publications:

“See our previous publications (Sanchez-Trigo et al., 2022; Scorcelletti et al., 2023) for more details on the results of the questionnaires on performance in the jumping and sprinting disciplines and the activity in metabolic unit per week, as well as the test results in this study on performance in the countermovement jump and forefoot hops.”

Given the different body sizes (Table 1) and your MR settings (260 axial images of 0.5 cm), were you able to scan the entire volume of all pelvic (Glutei) muscles for all individuals?

Yes, we recorded the complete volume of all gluteus muscles from all test subjects.

Do you have any insights into the accuracy of this semi-automatic measurement for assessing muscle volume? Please provide more details. Also, how long did the analysis take to quantify muscle volumes for this study?

The Mimics software provides an initial segmentation of all muscles in the form of so-called 3D masks. The final segmentation is then performed muscle by muscle and slice by slice in the form of manual corrections. The precision is therefore as good as with any other form of manual segmentation, i.e. between 1% and 2% depending on the size of the muscle.

The muscle segmentation tool takes about 30 minutes for the thigh and hip muscles and about 15 minutes for the lower leg. The duration of manual control depends on the number of muscles selected, the quality of the MR images and the automatically generated masks and, of course, on the training of the analyser. In total, between 2 and 3 hours are required for one subject and the 17 muscles selected here.

You forgot to mention the hamstring muscle volumes in your methods section.

Thank you for mentioning this. The hamstrings have now been added.

"Moreover, this selection.....joint stabilizers." It would be helpful for readers if you could specify which muscles you are referring to for locomotion and stabilizers (e.g., in brackets).

The sentence has been changed as follows:

Moreover, this selection included the major drivers for locomotion and some representants of joint stabilizers (m. sartorius, m. gracilis, m. tibialis anterior).

"We excluded three muscles....report any symptoms." Do you have any explanation for why those muscles showed such a large fat%?

We can only speculate about this. It is possible that these muscles are no longer properly innervated. Narici et al. (2004, see references) describe a generally reduced innervation of the musculature in old age. It could be that the extent of the loss of innervation in these 4 severely fatty muscles is particularly high by chance.

"The integrals of the water.....see Figure 3) as described....". Did the authors mean to refer to Figure 4?

Yes, of course. Thank you for mentioning this.

Were biopsies taken directly after the MRS measurement? Was there a specific reason why MRS was performed on the right calf and biopsies were taken from the left calf?

The biopsies were taken from the left calf, as a test for MVC and muscle elasticity was subsequently performed on the right calf, the results of which will be published together with the other biopsy data in a later manuscript (see above).

More information needs to be provided in Paragraph 4.4 to explain your analysis procedure in more detail.

These sentences have been added:

Using the manufacturer's software, the recorded force and the body mass determined at rest before the jump (weight / $9.81 \text{ m} \cdot \text{s}^{-2}$) were first used to calculate an acceleration value (ground force / body mass). From acceleration the speed of movement of the centre of mass was calculated by integration. The actual jumping power and the peak power reached before launch were then calculated from the product of the ground force and the speed.

Results

Ensure consistency in your decimal values (e.g., use 3 decimals throughout). Also, correct values such as 'P=844' to a standard format (e.g., $P=0.844$).

The journal requires the specification of three digits after the zeros, for example 0.00123 instead of just 0.001. $P < 0.001$ may only be written for values that are rounded to less than 0.001 (e.g. 0.000345).

At the request of the editor, I have also added a few things to the method section.

Of course, I have now also improved the $P=0.844$.

Where appropriate, mention percentage differences in the results section to give readers insight into the magnitude of differences. For example, indicate the percentage impact of age on the quadriceps

and hamstrings.

Concerning volume changes following has been added:

“Comparing pairwise by t-tests the sum volumes of the composed muscle groups of Masters athletes with the three other groups, we find for the glutei muscles a significant difference of +18% versus old controls ($P=0.0160$, $d=1.200$) and no significant differences to young athletes ($P=0.537$) and young controls ($P=0.115$). The volume of quadriceps muscle of Masters athletes differs by -18% from those of young athletes ($P=0.0269$, $d=-1.093$) and by +14% from old control ($P=0.0222$, $d=1.091$) with no significant difference to young controls ($P=0.840$). The volumes of hamstrings of Master athletes differ by +20 from volumes of old controls ($P=0.0132$, $d=1.275$) and were not significantly different from young athletes ($P=0.336$) and young controls ($P=0.149$). The volume of the triceps surae muscle of Masters athletes did not differ significantly from the volumes of young athletes ($P=0.691$), old control subjects ($P=0.306$) and young control subjects ($P=0.141$).”

Concerning changes in fat fraction following has been added:

“Comparing pairwise by t-tests the mean fat fraction of composed muscle groups of Masters athletes with the three other groups, the fat fraction of glutei of Masters athletes differ by +54% from young athletes ($P=0.00953$, $d=1.408$) and by -84% from old controls ($P=0.00351$, $d=-1.443$) with no significant difference to young controls ($P=0.577$). The fat fraction Masters athletes' quadriceps muscles was 43% higher than those of young athletes ($P=0.0046$, $d=1.576$) and was not significantly different from young controls ($P=0.0988$) and old controls ($P=0.131$). Concerning the hamstrings of Masters athletes, fat fraction was +57% higher compared with young athletes ($P=0.0490$, $d=1.012$) and not significantly different from old controls ($P=0.277$) and young controls ($P=0.266$). In triceps surae muscle, fat fraction of Masters athletes was +51% higher than in young athletes ($P=0.00133$, $d=1.966$), -34% lower than in old controls ($P=0.0392$, $d=-0.970$) and +33% higher than in young controls ($P=0.0242$, $d=1.115$).”

As stated in the legend of Figure 6, specify in the results section which muscles were included as 'jumping muscles'.

This has been added in brackets.

Discussion

The discussion largely focuses on muscle fat fraction, which is indeed an important aspect of the present study. However, it would benefit from a greater emphasis on how the current findings (both in muscle volume and fat fraction) align with existing knowledge.

A major and important aspect of the study is the differences in muscle volume between groups (and is also in the title of the manuscript). There is only limited discussion of these findings. For example, how does the data from the present study align with the literature (e.g., Fuchs et al. 2023; PMID: 36967049)? Compared to that study, sex, age, height, body mass, etc., are quite comparable.

Bringing your findings in light of current knowledge is important. Your study suggests that thigh muscles are more affected by age compared to other muscle groups, which seems consistent with previous work (Fuchs et al., 2023). Focus more on these aspects, as they form an important take-home message.

For example, irrespective of being active or not, thigh muscles seem to be the most impacted by age compared to other muscle groups. Of course, it needs to be acknowledged that there are limitations with a low sample size and the lack of detailed knowledge on the exact training regimen of all athletes (or please add more information on this in the methods, if available). Nevertheless, your data provides important further insight.

The first section of the discussion was also rewritten considering the comments of the other reviewer.

Masters athletes have significantly larger volumes in most thigh muscles and in the gluteus maximus muscle than the old control subjects. In contrast, no difference in muscle volume was found between the athletes and controls in the lower legs of the old test subjects. When comparing Masters athletes with young athletes in terms of muscle volume, a significantly lower volume of the quadriceps muscle was found in the Masters athletes. A recently published study on old and young people similar to our control subjects showed that in the old subjects a lower muscle volume was found mainly in the thigh muscles, less in the hip muscles and not in the lower leg muscles (Fuchs et al., 2023). In our study, this finding was only reproduced as a trend in the control subjects, which could presumably be due to the smaller number of cases in relation to the controls. However, if the controls and athletes are combined in our study, the earlier finding (Fuchs et al., 2023) that the thigh muscles in particular are affected by lower volumes with age is confirmed.

"...whereby the control subjects....activity." Add more details about this in the methods section. Also, in this sentence, change "do" to "did."

This sentence was deleted in the discussion. In the methods section, reference was made to an earlier publication from this study in which the physical activity levels are described in detail (see

above).

"The Masters athletes.... muscle volume." Although this may be true, it is based on a correlation, so there could be other factors at play that were not measured (e.g., neuromuscular factors). Consider toning down this statement.

The word "only" was exchange by "predominantly". If factors other than volume had played a significant role, the old control subjects and the Masters athletes would show a clearly different correlation curve.

"In contrast, earlier.....the IMCL..". This does not seem particularly relevant to your storyline. I would suggest focusing the discussion on the main findings and put them into perspective with age-related differences in muscle volume and fat fraction, rather than discussing acute (endurance) effects.

This sentence has been deleted.

Paragraph 5 ("It has previously....(Boesch & Kreis, 2000)."). What is the explanation why fat fraction is higher in the periphery compared to central part of the muscles? Please include more information on this.

The phenomenon of a heterogeneous distribution of fat in a muscle has been described several times without anyone having a mechanistic explanation for it. An increased accumulation of fat at the edge of a muscle could be related to interactions between the fat cells in the muscle and in the intermuscular fat. However, this assumption is so speculative that we do not wish to include it in the discussion.

Paragraph 6: Can the authors elaborate more on the findings regarding dorsal vs. proximal muscles? Apart from sitting, are there any other reasons that could explain these differences?

In this context, we only know the previously cited article by Grimm et al. The fact that you sit on the proximal-dorsal muscles can of course be pure coincidence and need not be the cause of the high fat content. We have therefore deleted the sentence referring to sitting. Other causes could be to be found in the areas of genetics and developmental biology. However, we can only speculate here.

Tables/Figures

Figures 1 & 2: Was there a reason why the authors chose to split the figure into two panels? As a suggestion, would the data be clearer if presented in four panels (A-D)? For example, one panel for hip muscles, two panels for thigh muscles, and one panel for lower leg muscles.

We have tried several variants. The existing variants with the division into two subfigures in two rows

proved to be the best compromise to show the data clearly and save space. Four figures in two rows with 2 graphs each would lead to narrower bars that would be more difficult to recognise. Four figures in four rows would take up considerably more space and would hardly be clearer than the existing variant. We would therefore like to leave Figures 1 and 2 as they are.

Figure 3: Would this data not be clearer if provided in a table? Consider presenting the data in a table format for better clarity and readability.

Figure 3 has been replaced by Table 4.

De Martino E, Hides J, Elliott JM, Hoggarth M, Zange J, Lindsay K, Debuse D, Winnard A, Beard D, Cook JA, Salomoni SE, Weber T, Scott J, Hodges PW & Caplan N. (2021). Lumbar muscle atrophy and increased relative intramuscular lipid concentration are not mitigated by daily artificial gravity after 60-day head-down tilt bed rest. *J Appl Physiol (1985)* **131**, 356-368.

De Martino E, Hides J, Elliott JM, Hoggarth MA, Zange J, Lindsay K, Debuse D, Winnard A, Beard D, Cook JA, Salomoni SE, Weber T, Scott J, Hodges PW & Caplan N. (2022). Intramuscular lipid concentration increased in localized regions of the lumbar muscles following 60 days bedrest. *Spine J* **22**, 616-628.

Drey M, Sieber CC, Degens H, McPhee J, Korhonen MT, Muller K, Ganse B & Rittweger J. (2016). Relation between muscle mass, motor units and type of training in master athletes. *Clin Physiol Funct Imaging* **36**, 70-76.

Fuchs CJ, Hermans WJH, Nyakayiru J, Weijzen MEG, Smeets JSJ, Aussieker T, Senden JM, Wodzig W, Snijders T, Verdijk LB & van Loon LJC. (2024a). Daily blood flow restriction does not preserve muscle mass and strength during 2 weeks of bed rest. *J Physiol*.

Fuchs CJ, Trommelen J, Weijzen MEG, Smeets JSJ, van Kranenburg J, Verdijk LB & van Loon LJC. (2024b). Becoming a World Champion Powerlifter at 71 Years of Age: It Is Never Too Late to Start Exercising. *Int J Sport Nutr Exerc Metab* **34**, 223-231.

Hawkins SA, Wiswell RA & Marcell TJ. (2003). Exercise and the master athlete--a model of successful aging? *J Gerontol A Biol Sci Med Sci* **58**, 1009-1011.

Joanisse S, Ashcroft S, Wilkinson DJ, Pollock RD, O'Brien KA, Phillips BE, Smith K, Lazarus NR, Harridge SDR, Atherton PJ & Philp A. (2020). High Levels of Physical Activity in Later Life Are Associated With Enhanced Markers of Mitochondrial Metabolism. *J Gerontol A Biol Sci Med Sci* **75**, 1481-1487.

McKendry J, Breen L, Shad BJ & Greig CA. (2018). Muscle morphology and performance in master athletes: A systematic review and meta-analyses. *Ageing Res Rev* **45**, 62-82.

Narici MV, Reeves ND, Morse CI & Maganaris CN. (2004). Muscular adaptations to resistance exercise in the elderly. *J MusculoskeletNeuronallInteract* **4**, 161-164.

Pollock RD, O'Brien KA, Daniels LJ, Nielsen KB, Rowlerson A, Duggal NA, Lazarus NR, Lord JM, Philp A & Harridge SDR. (2018). Properties of the vastus lateralis muscle in relation to age and physiological function in master cyclists aged 55-79 years. *Aging Cell* **17**.

Dear Dr Zange,

Re: JP-RP-2024-285665R1 "Leg and hip muscles show muscle specific effects of ageing and sport on muscle volume and fat fraction in Masters athletes" by Jochen Zange, Joachim Endres, Christoph Stephan Clemen, and Jörn Rittweger

Thank you for submitting your manuscript to The Journal of Physiology. It has been assessed by a Reviewing Editor and by 2 expert referees and we are pleased to tell you that it is acceptable for publication following satisfactory revision.

REVISION CHECKLIST:

We look forward to receiving your revised submission.

Yours sincerely,

Karyn Hamilton
Senior Editor
The Journal of Physiology

REQUIRED ITEMS

- Your manuscript must include a complete Additional Information section, including competing interests; funding; author contributions and acknowledgements.

- A Data Availability Statement is required for all papers reporting original data. This must be in the Additional Information section of the manuscript itself. It must have the paragraph heading 'Data Availability Statement'. All data supporting the results in the paper must be either: in the paper itself; uploaded as Supporting Information for Online Publication; or archived in an appropriate public repository. The statement needs to describe the availability or the absence of shared data. Authors must include in their statement: a link to the repository they have used, or a statement that it is available as Supporting Information; reference the data in the appropriate sections(s) of their manuscript; and cite the data they have shared in the References section. Whenever possible, the scripts and other artefacts used to generate the analyses presented in the paper should also be publicly archived. If sharing data compromises ethical standards or legal requirements then authors are not expected to share it, but must note this in their statement. For more information, see our Statistics Policy.

EDITOR COMMENTS

Reviewing Editor:

We thank the authors for replying to all reviewer comments. While the manuscript greatly improved already, there are some points raised by reviewer 2 which need further revision.

Senior Editor:

Thank you for submitting your revised manuscript for continued consideration in this Special Issue of The Journal of Physiology. Your revisions fully satisfied one of your Referee's previous concerns. However, your second Referee has some remaining feedback we'd like you to address. I believe it will require fairly minor revisions as most of the remaining feedback is geared toward enhancing the "story" told in the manuscript. We would like to again invite you to respond point-by-point with corresponding revisions to the manuscript. We look forward to seeing your revised work soon! Thank you again.

REFEREE COMMENTS

Referee #1:

This is an interesting study with some novel elements. In particular, the finding that masters athletes may compensate for age-related low volume of leg and hip muscles through intensive training could be of interest to researchers and practitioners. The authors have addressed my main concerns around the clarity of the narrative of lower muscle volume even in the masters athletes (compared with the younger athletes) and the updated text now better represents the study's reported results.

Referee #2:

Thank you for addressing the previous feedback and improving the manuscript. I have some additional suggestions and comments that I believe will further enhance the quality of the manuscript.

Abstract

The abstract can still be improved by showing relevant data. This directly provides the reader with some relevant insights into the data.

In the sentence: "In both age groups, athletes' muscles showed larger volumes than the corresponding control muscles", it would be nice if actual data can be shown.

You could consider adding the total amount of muscle volumes measured, or perhaps the sum of glutei, quadriceps and triceps surae (as this is also being specifically mentioned in the abstract, in the sentence before). Example: "In both age groups, the sum of glutei, quadriceps and triceps surae muscles showed larger volumes in athletes (young: xxx+xxx ml; old: xxx+xxx ml) compared to corresponding control groups (young: xxx+xxx; old: xx+xx ml) ($P < 0.001$).".

These data are also shown in Table 7 and the abstract figure. Therefore, this would nicely align and directly provide an important insight into actual data.

Subsequently, in the next sentence, it is relevant to zoom in on the comparison between athlete groups: "When comparing both athlete groups, only the volumes of the quadriceps and hamstrings... were smaller in the old compared to the young..".

Also, for fat fraction, can the authors provide some data here between brackets?

For example: "Fat fraction in the muscles of Masters athletes was higher (here the value) than young athletes (value; $P < 0.001$), lower than in old controls (value; $P < 0.001$) and comparable with young controls (value)".

This would also nicely align with the data shown in the abstract figure.

With adding a conclusion statement, my intention was that it would be useful for the reader to have an overall take home message/conclusion from your work, it was not meant to adjust the last sentence.

I would suggest leaving the last sentence as it was "This age-related... very small extent by the increased muscle fat content", followed by a concluding/take home statement. If there is no word room for it, you could leave it out. But, in my opinion, will make your abstract stronger.

Introduction

Overall, the introduction contains the relevant information related to your project. I have some minor suggestions (mostly repositioning of the sentences/parts) to improve the storyline. I believe this greatly improves the readability of your introduction and get to the main parts of your research question.

First of all, for clarity I would recommend to slightly adjust this sentence: "The age-related loss.....pronounced (Fuchs, 2023)." into "In the lower body, it appears that the age-related loss of muscle volume mainly affects the thigh(Fuchs 2023)". That makes a clearer link to talking about overall muscle volume (in the sentence before from janssen et al 2000) to the focus on the lower body, which is also what your study actually assessed.

Furthermore, I find that the introduction could use some improvement in the storyline (e.g. first the focus on muscle volume, then fat infiltration, then exercise/masters athletes). Most sentences/parts are good, but if it's a bit more organized this will strongly improve the readability.

To give some specific examples:

After "The age-related.... Fuchs et al. 2023)." It would make sense to add the paragraph about the impact of bedrest immobilization studies and muscle volume ("From bed rest immobilization..... (De Martino et al 2022)").

You could make a nice link by stating that the age-related loss of muscle volume could (at least in part) be attributed to disuse atrophy. and then state these findings "From bed rest immobilisation..... sartorius muscle (belavy et al 2009)."

This would provide a nice and clear section on the age impact on muscle volume.

Subsequently, you could put the focus on the fat infiltration (same principle: age-related findings first, followed by possible causes: disuse).

So in a separate paragraph you start your sentence: "Recent research further suggests concomitant accumulation of adipocytes.....". To introduce the fat infiltration.

This would provide a nice and clear section focused on the age impact on muscle fat infiltration.

Then you set the stage for mentioning the Masters athletes, and the (potential beneficial) impact of exercise training on these aspects (muscle volume and fat fraction).

So next paragraph: "Masters athletes are older people intensively".

This would nicely navigate the reader to the research aim(s): to assess the differences in muscle volume and fat fraction in older athletic individuals compared to older controls as well as the young.

Discussion

The discussion section would also benefit from improvement in the storyline. I would recommend to carefully look into the storyline to have clear sections throughout the discussion section to put your data in perspective. You don't need to change much on the sentences, but rather reposition it so you guide the reader better through your (nice!) data.

For example (as a suggestion):

For the introduction paragraph: Please start with only showing your main findings:

"In the present study, we observed...".

Then this could be followed, by

- (1) a paragraph focused on muscle volume differences.
- (2) then a section on findings and differences in fat fraction (IMCL/EMCL).
- (3) followed by the findings and section on countermovement jumps.
- (4) Finally, have the limitations paragraph and then the conclusions.

Such a storyline would also be in line with the present conclusion section as well as the key points and abstract: First the data on muscle volume, followed by the fat fraction data, and finally muscle function (countermovement jumping). This will provide a more logical structure for the reader.

END OF COMMENTS

Referee #2:

Thank you for addressing the previous feedback and improving the manuscript. I have some additional suggestions and comments that I believe will further enhance the quality of the manuscript.

Abstract

The abstract can still be improved by showing relevant data. This directly provides the reader with some relevant insights into the data.

In the sentence: "In both age groups, athletes' muscles showed larger volumes than the corresponding control muscles", it would be nice if actual data can be shown.

You could consider adding the total amount of muscle volumes measured, or perhaps the sum of glutei, quadriceps and triceps surae (as this is also being specifically mentioned in the abstract, in the sentence before). Example: "In both age groups, the sum of glutei, quadriceps and triceps surae muscles showed larger volumes in athletes (young: xxx+xxx ml; old: xxx+xxx ml) compared to corresponding control groups (young: xxx+xxx; old: xx+xx ml) ($P < 0.001$).".

These data are also shown in Table 7 and the abstract figure. Therefore, this would nicely align and directly provide an important insight into actual data.

Subsequently, in the next sentence, it is relevant to zoom in on the comparison between athlete groups: "When comparing both athlete groups, only the volumes of the quadriceps and hamstrings... were smaller in the old compared to the young".

Also, for fat fraction, can the authors provide some data here between brackets?

For example: "Fat fraction in the muscles of Masters athletes was higher (here the value) than young athletes (value; $P < 0.001$), lower than in old controls (value; $P < 0.001$) and comparable with young controls (value)".

This would also nicely align with the data shown in the abstract figure.

With adding a conclusion statement, my intention was that it would be useful for the reader to have an overall take home message/conclusion from your work, it was not meant to adjust the last sentence.

I would suggest leaving the last sentence as it was "This age-related... very small extent by the increased muscle fat content", followed by a concluding/take home statement. If there is no word room for it, you could leave it out. But, in my opinion, will make your abstract stronger.

We have carefully considered these suggestions. However, as the journal specifies a maximum length of 250 words for the abstract, we were only able to adopt the first two suggestions.

Introduction

Overall, the introduction contains the relevant information related to your project. I have some minor suggestions (mostly repositioning of the sentences/parts) to improve the storyline. I believe this greatly improves the readability of your introduction and get to the main parts of your research question.

First of all, for clarity I would recommend to slightly adjust this sentence: "The age-related loss.....pronounced (Fuchs, 2023)." into "In the lower body, it appears that the age-related loss of muscle volume mainly affects the thigh(Fuchs 2023)". That makes a clearer link to talking about overall muscle volume (in the sentence before from Janssen et al 2000) to the focus on the lower body, which is also what your study actually assessed.

Thank you for pointing this out. The sentence has been changed accordingly.

Furthermore, I find that the introduction could use some improvement in the storyline (e.g. first the focus on muscle volume, then fat infiltration, then exercise/masters athletes). Most sentences/parts are good, but if it's a bit more organized this will strongly improve the readability.

To give some specific examples:

After "The age-related.... Fuchs et al. 2023)." It would make sense to add the paragraph about the impact of bedrest immobilization studies and muscle volume ("From bed rest immobilization..... (De Martino et al 2022)").

You could make a nice link by stating that the age-related loss of muscle volume could (at least in part) be attributed to disuse atrophy. and then state these findings "From bed rest immobilisation..... sartorius muscle (Belavy et al 2009)."

This would provide a nice and clear section on the age impact on muscle volume.

Subsequently, you could put the focus on the fat infiltration (same principle: age-related findings first, followed by possible causes: disuse).

So in a separate paragraph you start your sentence: "Recent research further suggests concomitant accumulation of adipocytes.....". To introduce the fat infiltration.

This would provide a nice and clear section focused on the age impact on muscle fat infiltration.

Then you set the stage for mentioning the Masters athletes, and the (potential beneficial) impact of exercise training on these aspects (muscle volume and fat fraction).

So next paragraph: "Masters athletes are older people intensively".

This would nicely navigate the reader to the research aim(s): to assess the differences in muscle volume and fat fraction in older athletic individuals compared to older controls as well as the young.

We followed this suggestion and have further streamlined the storyline, whilst off with the principle idea of sarcopenia and atrophy as phenomena of muscle loss, and subsequently considering the influence of physical activity on both phenomena.

Discussion

The discussion section would also benefit from improvement in the storyline. I would recommend to carefully look into the storyline to have clear sections throughout the discussion section to put your data in perspective. You don't need to change much on the sentences, but rather reposition it so you guide the reader better through your (nice!) data.

For example (as a suggestion):

For the introduction paragraph: Please start with only showing your main findings:

"In the present study, we observed..."

Then this could be followed, by

- (1) a paragraph focused on muscle volume differences.
- (2) then a section on findings and differences in fat fraction (IMCL/EMCL).
- (3) followed by the findings and section on countermovement jumps.
- (4) Finally, have the limitations paragraph and then the conclusions.

Such a storyline would also be in line with the present conclusion section as well as the key points and abstract: First the data on muscle volume, followed by the fat fraction data, and finally muscle function (countermovement jumping). This will provide a more logical structure for the reader.

Thanks for pointing this out. We have now moved the paragraph on jump power in the Discussion section between the discussion on muscle fat and before the discussion of our model in Figure 5.

Dear Dr Zange,

Re: JP-RP-2025-285665R2 "Leg and hip muscles show muscle specific effects of ageing and sport on muscle volume and fat fraction in Masters athletes" by Jochen Zange, Joachim Endres, Christoph Stephan Clemen, and Jörn Rittweger

We are pleased to tell you that your paper has been accepted for publication in The Journal of Physiology.

Yours sincerely,

Karyn Hamilton
Senior Editor
The Journal of Physiology

If you would like to receive our 'Research Roundup', a monthly newsletter highlighting the cutting-edge research published in The Physiological Society's family of journals (The Journal of Physiology, Experimental Physiology, Physiological Reports, The Journal of Nutritional Physiology and The Journal of Precision Medicine: Health and Disease), please click this link, fill in your name and email address and select 'Research Roundup':
<https://www.physoc.org/journals-and-media/membernews>

- **TRANSPARENT PEER REVIEW POLICY:** To improve the transparency of its peer review process, The Journal of Physiology publishes online as supporting information the peer review history of all articles accepted for publication. Readers will have access to decision letters, including Editors' comments and referee reports, for each version of the manuscript as well as any author responses to peer review comments. Referees can decide whether or not they wish to be named on the peer review history document.
- You can help your research get the attention it deserves! Check out Wiley's free Promotion Guide for best-practice recommendations for promoting your work at: www.wileyauthors.com/eeo/guide. You can learn more about Wiley Editing Services which offers professional video, design, and writing services to create shareable video abstracts, infographics, conference posters, lay summaries, and research news stories for your research at: www.wileyauthors.com/eeo/promotion.
- **IMPORTANT NOTICE ABOUT OPEN ACCESS:** To assist authors whose funding agencies mandate public access to published research findings sooner than 12 months after publication, The Journal of Physiology allows authors to pay an Open Access (OA) fee to have their papers made freely available immediately on publication.

EDITOR COMMENTS

Reviewing Editor:

We congratulate the authors on a well improved manuscript.

Senior Editor:

Thank you for your revisions. We are now pleased to accept your manuscript for publication in The Journal of Physiology.
Thank you for your interest in The Journal and Congratulations!

REFEREE COMMENTS

Referee #2:

I have no further comments.